# Social Entropy and Normative Network

**DOI:** 10.3390/e22091051

**Published:** 2020-09-20

**Authors:** Emil Dinga, Cristina-Roxana Tănăsescu, Gabriela-Mariana Ionescu

**Affiliations:** 1Center for Financial and Monetary Research, Romanian Academy, “Victor Slăvescu”, 050711 Bucharest, Romania; emildinga2004@gmail.com; 2Faculty of Economics, Lucian Blaga” University, 550024 Sibiu, Romania; cristina.tanasescu@ulbsibiu.ro; 3School of Advanced Studies of the Romanian Academy (SCOSAAR), Romanian Academy, 010071 Bucharest, Romania

**Keywords:** social entropy, normative network, societal order, synergy, auto-poiesis, information, communication, predictability, uncertainty, hubbing, complexity

## Abstract

The paper introduces a new concept of social entropy and a new concept of social order, both based on the normative framework of society. From these two concepts, typologies (logical and historical) of societies are inferred and examined in their basic features. To these ends, some well-known concepts such as entropy, order, system, network, synergy, norm, autopoieticity, fetality, and complexity are revisited and placed into an integrated framework. The core body of this paper addresses the structure and the mechanism of social entropy, understood as an institutionally working counterpart of social order. Finally, this paper concludes that social entropy is an artefact, like society itself, and acts through people’s behavior.

## 1. Introduction

### 1.1. Aims of This Paper

The paper is aimed at delivering a new perspective on the concept of social entropy, which is related to the concept of social order and based on social justice precepts. To this end, three main issues are pursued: (a) A new comprehensive account of the concept of social entropy, especially for modern free and democratic societies; (b) a new comprehensive account of the concept of social order, especially grounded on social justice requirements; (c) two logical and inter-related models for social entropy and social order, respectively, as well as a co-evolutionary mechanism that connects the two models. Indeed, these two concepts are examined together and inter-conditioned—as a paired item. To this end, a set of related must be defined and integrated into the intended comprehensive account of the social entropy—including network, synergy, order, system, norm, autopoieticity, fetality, hubbing, and complexity. Our hope is to help orient the research on social entropy toward a direction that strictly links this concept to its institutional and cultural paradigm, by removing it from the dangerous and unproductive concepts of mechanic-ism and thermodynamic-ism.

Indeed, this research is engaged in answering at least four questions: (a) What is social order? (b) what is social entropy? (c) how are social order and social entropy inter-conditioned and what is the mechanism of such inter-conditioning? and (d) what is the social justice background that social entropy could be modelled on?

### 1.2. Methodology

The basic methodological feature of this research is a logical analysis. In addition to this analysis, an institutional examination is also focused on concepts describing social order as well as those describing social entropy. Indeed, the main methodological lines pursued are the following:Putting the entire analysis under a conceptual social umbrella—that is, the physicalist account of the concept of entropy will be rejected from the start, as simply an uncritical import from Thermodynamics;Endowing the analysis, interpretations, and results with (as much as possible) an abstract to apply them to particular empiric cases;Using a combination of lexicographic and logical criteria when some rankings of concepts or processes involved in the research must be carried out;When appropriate, the definitions are performed based on a minimal list of sufficient predicates (attributes) to provide, as much as possible, applicability to particular or local cases;Generally, we appeal to an intuitionist perspective when some concepts, mechanisms, and specific methodologies needed to be proposed and examined;When possible and, to offer more clarity, completeness, and suggestiveness, some logical formalisms are delivered, using the standard logical constants;The study also appeals (for the most subtle cases) to graphical representations, which have the potential to provide synoptic insights and deliver additional arguments for the ideas under analysis.

### 1.3. General Organization of the Paper

This research starts with a background that introduces and explains our insight into the main issue to be examined, including our own typology of the lanes through which the concept of entropy entered the social field from Thermodynamics. Thus, basic concepts such as system, network, social system, normative framework, normative network, fetality of norms, and complexity, are introduced and prepared to be used in the core body of the research.

The first concepts approached to ground social entropy are order and social order, respectively, which are analyzed from both common and intellectual perspectives, by describing their mechanisms of appearing, revealing, recognizing, and changing. Social order is modelled as DTI model (that is, a model that contains the following three components: Self-respect, freedom, democracy).

Then the concept of social entropy is introduced, which is closely connected to social order. Social entropy is taken far away from its technical (inappropriate) features from Thermodynamics and is approached as a social artifact (like social order). Social entropy is modelled as DIC model (that is, a model that contains the following three components: Negative discrimination, economic inequality, corruption).

This study advances its own position concerning the concept of complexity, which connected with the artefactual origins of social order and social entropy.

Based on the concept of social entropy, this study proposes its own typology of modern societies (the IIHH model of societal typology—that is, infected, ill, healing, and healthy societies, respectively).

The concepts of both social order and social entropy are used further to propose (as a sketch) an autopoietic model of modern society.

Finally, this paper identifies four scientific topics to be continued/followed/initiated in the margins of the discussion and the results obtained to date in the topic of social entropy.

### 1.4. Main Contributions of Our Research

This research makes the following contributions (both theoretical and methodological): (a) “moving” the concept of social entropy from its mechanic-ism and bizarre analogies with Thermodynamics into the territory of the social field; (b) connecting, causally, structurally, and functionally, social entropy with social order; (c) providing operational definitions for social order and social entropy based on the their quality as artefacts; (d) proposing logical models for social order (DIT model) and social entropy (DIC model) and their connections into a social abstract mechanism; (e) considering social justice as the foundation of both social entropy and social order, thereby moving towards a moralized social contract; (f) proposing an autopoietic model of society, based on social entropy and social order; (g) proposing a typology of present day societies based on the criterion of social entropy.

## 2. Background

### 2.1. Some Current Ways of Using the Concept of Entropy in Social Sciences, as Related to Paper’s Topic

The original technical concept of entropy is well-known. There is already extremely rich literature on the primary concept of entropy (originating in Thermodynamics); consequently, this concept is well-known to everyone, and a fortiori, to the readers of this publication, so our task is considerably eased in introducing the concept. Thus, we will highlight only the main attributes that are usually associated with the primary concept of entropy: (a) Entropy is a concept of maximum generality, which is equally applicable (in essence) to any of the three Popper-ian worlds, as well as to combinations of them, just like the social system; (b) it has three meanings (ontological—as the parameter of a real entity, objective or subjective; epistemological—with cognitive significance for the subject (indicating the level of the order that the entity experiences); and methodological—as a selector of the procedure for ”querying” the object); (c) its referent (denoted) is the state of an existing entity (system or process); thus, entropy can be formalized as a state-variable, a state-function, or a state-vector; (d) the size of entropy’s variation does not depend on the intermediate stages (that is, on the ”road” pursued), but only on the initial and final points; (e) the state referred to by the concept of entropy is embedded by the concept of order; (entropy is a parameter that ”moves” inversely in relation to the associated order; more precisely: the size of the entropy is inversely proportional to the degree of its ordering; (f) entropy is non-static: in a closed system (e.g., our Universe) entropy necessarily and permanently increases; (g) thus, global entropy (that is, entropy inside a closed system) is irreversible; (h) entropy is a macroscopic variable, determined by modelling the integration of microscopic states. Thus, entropy signifies macroscopic irreversibility derived from microscopic reversibility (see, also, the problem of Maxwell’s demon); (i) entropy is a statistical quantity (based on the statistical formulation of Thermodynamics)—this justifies the appearance of probability in the analytical formula of entropy in Statistical Thermodynamics, because probabilities can only model the average of a population; (j) entropy is an additive variable. Indeed, with the concept of entropy, whose origin is claimed by Thermodynamics in trying to escape from micro states (imposed by Mechanical Statistics) and move towards macro ones, importation of the concept occurred like with most of the basic concepts (and, often, principles) in the social sciences (primarily, Economics): An uncritical and tale quale import from the natural sciences. This appetite, which is difficult to understand, of social sciences for foreign quantitative and a-contextual conceptual constructions has not yet tired. In the present paper, we will try to depart as far as possible from such an undesirable epistemological account.

There are three apparent strands (this typology is our proposal) of importing (not constructing) the concept of entropy into social sciences: (a) A strand based on *similarity*—that uses a Procust-ian gathering of the concepts imported to isomorphically find the Thermodynamical design of entropy, including temperature, energy, exergy, etc. [1] (especially from an economic perspective (Tribus is credited with coining the term Thermoeconomics)). Such an approach remains at the intersection between Physics and Economics; (b) a strand based on *analogy*—where the original concept of entropy is only a starting point, instead of an analogous model of functioning in spirit but not in letter built to attain an entropic understanding of the social (in fact, of the economic) process functioning [2]; such an approach is located deeply in the economic/social process albeit with a too little appeal, in our opinion, to sociality and the specific third Pooper-ian world that subsists inside the social system; (a very productive concept here is fund, which is a stock with only outputs but no inputs (a pure example is the Sun); the opposition between stock and fund is an interesting factor to understand the entropic functionality of society as a whole). (c) A strand based on *operational sociality*—our own model of society’s structural functioning, which constitutes, in our opinion, a promising way to build, not to import, a genuine concept of social entropy [3]; this time, in the specialty literature, five variables, typically social, are identified and embedded in a logical structural model known as PISTOL/PILOTS (population, information, space, technology, organization, and living). It is clear why we would name such an approach as “operational sociality” and not, for example, a pure or full sociality approach. What we propose in this paper is a fourth strand of approaching the concept of social entropy, which can be appropriately named *social justice-based social entropy*. Since Bailey’s work [3] seems to be the closest to our approach, some of our positions related to his SET model (social entropy theory) proposal are listed below:(a)SET proposes a benchmark of maximum social entropy, against which the actual level of social entropy can be measured; instead, we accept a gap of actual entropy against the previous level of social entropy, because we cannot know the maximum level of social entropy;(b)SET is still strongly linked to the information base of entropy, exactly through the variable named “information” in the PISTOL model of social entropy, while our DIC model of social entropy does not appeal to information in its understanding of information theory;(c)Among the eight components of the social system, many are non-independent from one other (e.g., culture and information, and energy and technology); in our approach, independence and consistency among the DIC components are ensured;(d)Generally, the PISTOL model is maintained within a Thermodynamic “tunnel”, because this work institutes a membrane as a filter for inputs (and outputs) in the external relationships of the social system with its environment. In our proposal, filtering is performed by the behavior itself, within the social system.

### 2.2. System and Network

There are some important differences between systems and networks, which often ground basic reasoning and put limits on, or trigger properties of, functioning of the two conceptual entities. Since our paper will ground social entropy in a normative network, we briefly deal with this latter issue.

Generally, a system is an intellectual division based on reality (objective, subjective, or subjectively objectified) using a membrane—such a membrane is, in the last instance, cognitive but, in intermediate instances, it can be physical, institutional. or can remain cognitive. This is the first stage of the observation—the *distinction*—which is followed by the second one—the *indication* [4].} The relationships among intra-membrane elements generate the system *functionality*, while the relationships between the intra-membrane elements and the environment generate the system *behavior*. The difference between any aggregate of elements and a system consists of splitting the intra-membrane elements into two categories: (a) The leading sub-system, and (b) the executive sub-system. The leading sub-system has orders/commands to the executive sub-system (which are inputs for the executive sub-system) as outputs and uses reports on the execution of order/commands from the executive sub-system (which are outputs of the executive sub-system) as inputs. Thus, a system is always a role-based heterogeneous entity (no matter how many levels of hierarchy there are, if there is at least one level (as discussed), then it is a sufficient predicate to qualify that entity as a system), with a sub-system occupying a privileged place—the leading one—that generates vertical-oriented interactions between the component elements, alongside the horizontal-oriented ones.

### 2.3. Social System and Network

A network is a system that has lost its role-based heterogeneity. We do not believe that a network always originates in a system that loses its leading sub-system. Our definition of “network” uses the Aristotle-ian definition of definition, which is logical, not causal or generative. Although different theories of the social system focus (as their *raison d’être*) on different ends (for example, the mutual advantage of cooperation, or the reciprocity among society’s members), the common denominator of all approaches (either explicitly or implicitly) is synergy. Throughout our paper, the social system will be understood as a machine that generates synergies of any kind. Likely, the phenotype of the social system is neither the individual (see Maturana) [5], nor communication as such (see Luhmann) [6], but rather social synergy. Based on this insight, one could build a comprehensive theory of the social system’s autopoieticity. Nonetheless, it should not be understood that synergy only addresses the addition of new properties, as synergy can also fully delete or replace existing properties with others. For example, the competition for resources, in the state of nature, is replaced by cooperation under the social contract in the state of society. The same is true for the Kantian reciprocity, which is an absolute additional property, that cannot be found in the state of nature.

Modern societies are free and democratic social entities. Based on the social contract, any society cannot be more than a system, that is, it must have a leading sub-system (generically named a political sub-system) and an executive one (generically named civil society). We especially consider the contractarian theory of Rawls [7], to which some subtle adjustments from Robert Nozick, Amartya Sen, and Martha Nussbaum [8] we implicitly recognize. The democratic character of society is objectified through political public decision making (Parliament and Government, based on the rule of majority, as holders of the sovereignty of people), while the free character of society is objectified through a set of fundamental liberties—of moving, thought, expressions, associating, voting, consciousness, etc. By combining the two pillars of human society, we can derive the following logical and institutional consequences.

(1)Society is a system of systems, that is, a supra-system: In the general social system, other systems are included, each of them verifying the basic sufficient predicates of a system—leading to role-based heterogeneity. Society cannot be a network, a network of systems, or a network of networks, because, as noted, the social contract is in force. This social contract necessarily requires a leading sub-system.Inside any system there is at least one network. Such an assertion is justified (required) by the fact that the social system must verify the predicate of freedom, which implies that it must contain at least one network, given that freedom is inconsistent with vertical decisions for individual ends (see the works of libertarians, such as Hayek [9] and others within the Austrian economic school). This “theorem” is crucial for modern societies in which at least two sub-systems function as networks: The economic sub-system, and the cultural one. Freedom is the ”first” primary good [7], while democracy itself is the ”secondary” primary good. Indeed, it can be shown that democracy is logically inferable from freedom, but freedom is not logically inferable from democracy.(2)Any network is auto-poietic; the sufficient predicates of the autopoieticity are: (a) Autonomy of internal operations (operational closing) (we remind the reader that a network, like a system, has a membrane that differentiates it from the environment; thus, internal operations are operations inside the membrane). (b) roughly identical replicability of its phenotype, and (c) co-evolution with its environment, inside the “tunnel” of identity conservation (inter-actional opening). Autopoieticity was coherently introduced in the literature by Humberto Maturana and Francisco Varella [5], for the biological living systems. For its extension to social field, it seems German sociological Niklas Luhmann went further (than others) to enough articulate a theory of the social system based on the autopoietic property of a system [6]. Autopoieticity and entropy are very strong connected, so we will later address this interesting property further. Although Luhmann draws on Maturana, he still remains within Parsons-ian functionalism. In fact, Luhmann experiences some eclecticism: He is structuralist when rejecting people from the social system, but he is functionalist when holding communication as the phenotype of the social system.(3)Together with systems of systems, networks of systems, systems of networks can exist, as well as networks of networks.(4)Nonetheless, no networks, networks of systems, or networks of networks can exist outside a system, although a system can exist outside a network.

While systems can be hierarchized (either by inclusion or based on other criteria of interest), networks can never be hierarchized as such. In other words, systems can be relatively positioned either vertically or horizontally to each other, but networks cannot be positioned other than horizontally to each other. A question arises here: Networks, like systems, can be included within each other. However, in such a case, does this not become hierarchically positioned over the included network? The answer is negative: Since semes can never be transmitted between networks, they can never be hierarchized among them. The most convenient example is civil society (i.e., the typical social network) all of whose components are related horizontally only to each other.

These six characteristics of the social system constitute the conceptual grid inside which the discussion of the relationship between social entropy and the normative network will be carried out throughout this paper.

### 2.4. The Normative Framework

The normative framework is an oblique process inside the social system, because it is both vertical (since there are norms enacted by the leading sub-system for itself, as well as for the executive sub-system) and horizontal, either inside the leading sub-system or the executive one.

(a)The vertical path of the normative framework:From the leading sub-system emerge *semes*, both for itself and for the executive sub-system. We define semes as norms with the following attributes: (i) Explicit; (ii) formal; and (iii) publicly enacted. For example, an explicit and formal norm enacted by a private organization is not a seme. Semes can be both discretionary and automatic. Automatic semes must be discretionarily implemented but they must further work automatically. The term seme is based on the Greek *sēma* (sign) but we assign the above triangular meaning. Here, we are approaching Luhmann’s concept of communication (as a phenotype of the autopoietic social system) but, in contrast with Luhmann’s account, communication (the seme) is only a functional “property” of people—more precisely, of the people included in the leading sub-system. The term “seme” is also used in semiotics (more precisely, in semantics) where it represents the smallest unit of meaning of a sememe/morpheme. Here and elsewhere we use the term “seme” with the meaning proposed above, that is, as a species of norm.(b)The horizontal path of the normative framework:Among non-hierarchized systems the normed relationships are carried out through *memes*. The meme is a species of norm that holds the negated attributes of the seme (that is, it is implicit, informal, and not publicly enacted) [10]. A meme is never enacted as such, it is just contained in objectified behavior from which it is extracted by imitators.

Of course, the distinction between seme and meme is simply one of theoretical and methodological interest, since, in reality, many useful memes are codified over time, and thereby become semes, while some semes that are proved to be unhelpful are rejected from normative codes and become, at most, memes, if they do not disappear. The ventilation between semes and memes realms is a praxiological effect of the social system real functionality. However, what are useful memes? Such a predicate is gained when, for a given meme (or a homogenous group of memes), its use disjunctively verifies two conditions: (a) It quantitatively passes beyond a critical threshold, and (b) it qualitatively substitutes, to a significant degree, the analogous seme. Moreover, what institution could/should monitor the emergence of useful memes? Since the normative framework of society is a direct result of the social contract, that is, of constitutional principles, the “guardian” of the memetic processes must be the Constitutional Court, which should present reports on the normative framework functioning in Parliament biennially or, even at the beginning of every legislature. Consider the appropriate way to design an alley in a park: For a while, people will be allowed to cross the park where they want; then, the alley will be established on the busiest route.

### 2.5. Norms Fetality and the Normative Network

Norms are a necessary outcome of the social contract, from both a contractarian or procedural-based view and from a result-oriented one. Based on the political principles designed when the social contract is made (that is, in the original position [7]), norms are stipulated into normative acts (first through the Constitution, based on which institutive laws. i.e., primary legislation, and procedural laws, i.e., as secondary legislation, are enacted and implemented based on the state monopoly to sanction the violation of the concerned norms).

The role of these norms (and, more generally, of the normative framework of society) is thus to build sufficient social channels of interactions so that synergy can arise and replicate indefinitely.

The fetality of norms means that the features of norms will be consistent, given a pre-existing normative background, with the norms already working in society, similar to graft fetality in the biological realm. To some extent, any new semes are necessary because of their “pre-existence” in the normative axioms or principles (hosted by the Constitution). They are also necessarily consistent with those axioms; however, other cases could happen, when such consistency reaches its limits of acceptability or even goes outside them (for example, revolutionary cases). The fetality of norms is a very important issue when one wants to discuss social entropy, as social order can be disturbed not only by lack of norms (or their insufficiency), but also by an abundance of non-fetal norms. In a certain sense, the fetality condition is a filter for the internal order of society analogous with the concept of “normal science” [11]. It is also analogous with Luhmann’ programs to generate communications [6], or Lakatos’s research program [12]. As a filter for desirable norms, the fetality avoids the particular noise of norms, which is understood as non-fetality. Since fetality, viewed at the whole normative framework level, bears much of what we will propose further to be understood by the social entropy, we provide further commentary on this matter.

(a)A non-fetal norm verifies the following inclusive logical disjunction:
$${\bar{F}}_{i}^{NF}=\left({\bar{C}}_{i}^{NF}\right)⋁\left({\bar{P}}_{i}^{NF}\right)⋁\left[\left({\bar{C}}_{i}^{NF}\right)⋀\left({\bar{P}}_{i}^{NF}\right)\right],$$
where F signifies fetality, i counts the time of examination, C signifies the property of consistency with the current normative framework (NF), and P signifies the property of convergence with the general end of society grounded in the current normative framework ($\bar{x}$ signifies the logical negation of x);(b)A fetal norm is the logical negation of a non-fetal one, that is: $F_{i}^{NF}=\left(C_{i}^{NF}\right)⋀\left(P_{i}^{NF}\right)$; indexing the properties with time (i) is very important, because the properties are strongly contextualized. Of course, the context of the normative framework is not so variable that it changes every year. For example, generally, a core of normative principles (usually in the Constitution) are invariant for a long period, which is why the index “i” is used, at least in empirical analyses, for representing a number of years (e.g., five years), after the case;(c)Based on the fetality property of norms, the normative evolution of society can be classified as: (a) Conservative evolution—which verifies a high degree of fetality; (b) innovative evolution—which verifies a low degree of fetality.

As we already argued, the normative framework has an oblique shape, so it contains both semes and memes. By combining the fetality of norms with the normative framework we can obtain the concept of the normative network, which is the second term in the equation *social entropy–normative network*, which our paper aims to “solve” (the first one being social entropy).

Firstly, we need to apply the concept of the autopoieticity of the social system, in general, along with its networked components. One of the sufficient attributes of an autopoietic entity is, as stated before, its operational closing, i.e., the property that any interactions within the given membrane are driven internally. Nonetheless, there are always external norms that emerge from the leading sub-system that embed themselves in the concerned autopoietic entity, which will tend to open that operational set.

Secondly, we must combine the autopoieticity of the entity with the fetality of the external norms. If the external norms are fetal for the entity concerned, then they will be internalized, and their external origin will be “forgotten” and fully replaced with an internal one. This, however, does not mean that a leading sub-system will arise inside the entity concerned, but simply that those norms acquire “citizenship” to the entity. Next, we propose a mechanism to derive a normative network within a social system. We also explain why we need a normative network and why it is not sufficient to handle the normative framework.

The evoked mechanism used to generate the normative network is synoptically shown in Figure 1.

## 3. Social Entropy

### 3.1. Social Order

#### 3.1.1. The Common Way to Establish the Order

Common language has, of course, a definition for the concept of order, which is considered a configuration, spatial, temporal or (most often) a combination of the two, that characterizes a phenomenon and is detectable as being such a configuration, i.e., it is intelligible to the empirical observer, without presuming any special powers or competences of the observer. In other words, the “common” order is simply a pattern, static or kinematic, which the ordinary observer “throws” over a real phenomenon (objective, subjective, or objectified) and that fits the reality in question. Two questions are important at this point: (a) What is the “method” by which the pattern in question is detected? (b) What is the potential to make (true) predictions for this pattern.

(a)Clearly, in the common sense of the term, the order observed at the non-specialized empirical level is of an inductive type. The observer finds regularities, periodicities, and other characteristics of the replicability of a phenomenon and, by inductive inference, i.e., by generalization, s/he constructs the order in question. Of course, inductive inference is liable to produce falsehoods, (as many logicians have shown (including Bertrand Russell—see Russell’s hen—and Carl Hempel—see Hempel’s raven) so the common concept of order is a vulnerable one; deductive inference is no less susceptible to falsification, but it is a logically different type of falsification (see Popper’s factual falsifiability).(b)In terms of the potential to allow the formulation of true predictions, the common concept of order is extremely deficient. Of course, predictions can be made, but they are true (i.e., the previous predictive statement coincides with the subsequent descriptive statement) only by chance. Here we discuss correspondence-truth, credited to Popper’s theory of factual falsifiability. The explanation is that generalization has an enormous probability of being local (both temporally and spatially), while the truth of a prediction is “governed” by the universal character of the major premise, not by its general character. Universality (at least hypothetically) is ensured by deductive inference only. Here, moreover, exists the logical difference between inductive and deductive truth. In the case of inductive truth, the major premise of a syllogism is a general coverage law, while in the case of deductive truth, the major premise of the syllogism is a universal coverage law. Therefore, the commonly accepted concept of order consists of considering reality as it appears to the non-specialized observer and inferring the causality (i.e., the order’s patterns) by generalization, i.e., by induction. We note, for the sake of intellectual honesty, that all scientific knowledge uses induction not in its non-critical form, which we have discussed above, but in the form of abduction i.e., the most plausible explanation, prima facie, inductively obtained but subsequently subject to a deductive mechanism.

#### 3.1.2. A Logical Way to Establish the Order

Clearly, from a scientific perspective (firstly, from an epistemological perspective), we need a different examination of the concept of order. This process consists of identifying the predicates of sufficiency that the concept of order must verify to be qualified. We propose these sufficient predicates as follows:

(PSO1) *sensory observability*: The phenomenon that will underlie the discovery (or non-discovery) of an ordered configuration, which must be observable at the sensory level, even if, a sensory sensation, perception, conceptualization, and judgment (reasoning) ultimately lead to the formulation of the order in question, by universalization; from a gnoseological (but not ontological!) perspective, not finding an order is equivalent to the non-existence of order, because order exists only in association with a subject-observer. From this point of view, Phenomenology is very close to the opinion expressed here.

(PSO2) *intellectual observability*: If PSO1 involves the natural senses of the cognitive subject, then PSO2 involves the intellect of that subject. Perception is the form that consciousness gives to sensation. To form concepts and prepare judgments, the intellect of the subject is needed, which transforms perceptions into concepts. Conceptualization is the most crucial stage in the deductive identification of order. It should be mentioned that sensory observability is common to non-cultural and cultural subjects, even if the former cannot achieve the concept of order. However, from an empirical point of view (and especially with the help of memetic learning), non-cultural subjects can access a certain static (and, sometimes, even kinematic) “map” of the space and time interval in which they lead their lives.

(PSO3) *catalogue registration*: The application of the first two predicates of sufficiency only leads to the possibility of notifying an order. The actual notification of the order occurs only if the result of the application of PSO2 is found in the pre-existing catalogue of possible orders, a catalogue that is accessible to the cognitive subject in question.

The order (O) can be given by the logical conjunctions of the three predicates of sufficiency:A ← (PSO1)⋀(PSO2)⋀(PSO3)

A short discussion of this concept follows:(i)How does the cognitive subject come into possession of the catalogue that contains the list of already known orders? This can occur only as a result of previous experience, so this catalogue is always an a posteriori one;(ii)How is the new possible order actually compared with the catalogue records? Here, one could postulate a fourth predicate of sufficiency to guarantee this operational capacity. We consider, however, that this new presumptive predicate would be redundant, so we instead assume that the cognitive subject holds the needed intellectual capacity in this matter on the simple basis of the quality of the cultural subject;(iii)If the cognitive subject does not find the order in the catalogue, how does s/he proceed? There are two alternative options: (iii.1.) s/he “decrees” that there is no order in the phenomenon in question; here a type 1 error exists—the rejection of a true hypothesis; (iii.2.) s/he completes the catalogue with the new presumed order; here a type 2 error exists—the admission of a false hypothesis. From a logical point of view, if we do not accept the a priori nature of the catalogue of orders (as already assumed above), the catalogue will be formed, over time, for each cognitive subject separately, exclusively based on the practice of type 2 errors;(iv)Are there also catalogues of orders that are supra-subjective (supra-individual)? The answer is obviously affirmative: In society, there is a common catalogue (social and communitarian) of orders, which is shaped by democratic “votes”;(v)Both the catalogues of the orders of individuals and those of the community are, in turn, established by fields of interest or typological fields, including ontological orders, gnoseological orders, axiological orders, praxiological orders, etc. A summary of all considerations regarding the concept of order from a logical perspective is provided, from a synoptic point of view, in Figure 2.

### 3.2. Revealing the Order

Revealing the order refers to gaining conviction, at the level of the empirical observer, that in the entity that plays as object of the observation, there is an order. At least three questions result from this method of defining the revealing of the order: (a) What happens if the order is not revealed? (b) What happens if the empirical observation reveals a single order? (c) What happens if the empirical observation reveals more than one order? We will examine these questions one by one.

(a)The disclosure does not indicate any order.If the result of the empirical observation does not overlap with any existing order in the list of orders, then the subject will “decree” that there is no order. The non-existence of an order does not prohibit the praxeology inside the entity in question, but this praxeology will either be operationalized “blindly” or will be based on the imposition of a sui generis order, generated by the operationalized praxeology itself. The second alternative—imposing a sui generis order—will introduce a new order in the list of orders, by way of a type 1 error. We call praxiological operation “blinded” when actions are not based on known causalities (determinisms or statistical regularities).(b)The disclosure indicates a single order.This is the standard case, in which the cognitive (and praxiological) subject identifies an overlap between the empirically observed order and a record (and only one) in the previous list of orders. This is a conservative and unproblematic case;(c)The disclosure indicates more than one order.To find more records in the order list following empirical observation is much more problematic than the two cases previously discussed, for the following reasons:How can the order that will govern the praxiological action be “chosen”? A situation in which the praxiological action will take place within two or more orders simultaneously is not acceptable, because each order has, as we have shown above, its “own logic”. Operating under several orders at the same time not only introduces incoherencies (or even inconsistencies) into action, but can also evade rationality as such, given that a certain order is unequivocally associated with a certain model of rationality. In this situation, the concept of adequacy is useful. This concept has a vagueness generated by its use in the common (“civil”) language so we will have to give it a more rigorous meaning. We propose that, by praxiological adequacy of the revealed order, it is understood that property of the order in question is located at the smallest “logical distance” from the intrinsic logic involved in the intended praxiological action. Notably, the adequacy of the revealed order does not refer to the adequacy of this order for the purpose of the praxiological action involved, but to the “mean” of that action, i.e., to the intrinsic logic of the action in question. It is pedantic (and useless) to formulate models for “choosing” the right order, because the real actor will never use such models (either because s/he does not know them or because the transaction cost for their use is prohibitive). We believe that each actor will determine, on an intuitive basis, but especially using past experiences, which order among those revealed to her/him is the most appropriate;(d)Could the lexicographic ordering of pre-known orders from the list of orders available to the observer, provide a criterion for choosing the most appropriate order? We think that the answer here should be negative. Lexicographical ordering (i.e., one without a criterion) will be of no use, because each praxiological situation features its own descriptions of intrinsic logic which are, as proposed above, decisive in choosing the most appropriate order.

### 3.3. Order and Entropy

Based on its features, entropy can be understood as an orderer of reality. The most effective proxy in perceiving/identifying the order of reality seems to be the structure of the intentionally targeted entity. There are two categories of primitive, ontological structures of order: (a) A *causal* structure, “responsible” for explanations/predictions and for altering the identity of the entity, and (b) a *coexistence* structure, “responsible” for functions/outcomes and for preserving the entity’s identity.

The preferential meaning of change in the entities of objective reality is given by necessity, which is the only “arrow” of finality. Necessity is “designed” exclusively by physical laws (biological or chemical “laws” that are ultimately reducible to considerations of quantum physics). Except, of course, in the case of living reality, where a materialist reductionism is inapplicable.

Based on the concept of entropy and the entropic mechanism, we next examine the relationships that could exist between entropy and order.

According to its significance in Thermodynamics, entropy means a tendency towards the homogeneity and non-differentiation of a process or system. In this sense, the idea that an increase in entropy indicates an increase in disorder is risky, as any spatio-temporal, causal or functional configuration, can signify an order. For example, financial stability is perceived as indicating a high degree of economic order, although it is characterized by a greater homogeneity of the financial process (by reducing the number or the amplitude of monetary shocks or volatilities);In our opinion, the use of entropy for the identification, “measurement”, or evaluation of order must be done with great caution, which we will strive to do; the tale quale import of the concept from Thermodynamics can be especially dangerous;In the case of dissipative systems, the concept of Thermodynamic entropy can play a role, but likely only in terms of an exotic label for phenomena/events that already either have their own associated terms or can be named without appealing to the concept of Thermodynamic entropy.

### 3.4. Social Order

To define the concept of social entropy we must first build the concept of social order, which is, as shown, its quid pro quo. In fact, entropy and order are opportunity costs for each other. Any society lives based on a social contract, that is, based on principles that are, usually, are cast into laws—or, more generally, into norms. Thus, finding the social order means, somewhat, *chercher les règles*. Some preliminary will smooth the road towards this concept:

We think that there are three pillars upon which a society is built: (a) *Self-respect* (or moral dignity), i.e., mutually recognizing and defending the human condition at an individual level; (b) *freedom* i.e., mutually avoiding to imposing one’s private goals upon others; (c) *democracy* i.e., the actuation of the majority’s goals and means through socially made choices. For the last pillar, there is a sociological account according to which the minority tends (almost necessarily) to impose its goals and means over the majority, based only on the desire of the majority not to negatively discriminate against the minority—a recent exploration of this idea can be found in Taleb [13]. Some philosophers (for example, Rawls) define self-respect as the social basis for justice (i.e., social justice), but we believe that it can express more than that, i.e., the whole social order.

Discussion: (a) We argue that the most appropriate lexicographic descending order of the three pillars, according to their natural justification, is (a), (b), (c); (b) a synoptic illustration of the relative positioning of the three pillars is shown in Figure 3. The fact that freedom is not the fundamental pillar in society might be quite polemical, but history is imbued with examples that prove our ordering—see the saints or the heroes; see also Carlyle’s conceptual distinction between eleutheromany and gigmany [14]. In fact, we propose here a lexicographical ranking of the three pillars, in the same way that libertarians do (and even social philosophers, such as Rawls) placing freedom in first place.

(i)The functions in Figure 3 are as follows:
(i.1)The maximal social order is reached when the relationship between freedom and democracy is such that the self-respect of individuals in that society is maximized (point M);(i.2)If democracy has “negative” values (to the left of the vertical axis), then society becomes anarchical;(i.3)However, if the democracy is too analytical, by regulating almost all social interactions from the perspective of the majority, the result is a dictatorship (to the right of the point N);(i.4)Both freedom and democracy cannot experiment with all theoretical degrees, because they could then become impracticable (to the left or right of the minimal or maximal degree values of freedom and/or democracy);(i.5)Interestingly, that under a certain degree of self-respect, society enters animality, while over a certain degree, it enters selfishness;
(ii)The concrete point at which the self-respect of the individual reaches its maximal level is not theoretically identifiable, but can only be determined empirically (that is, socio-historically contextualized), because it depends from the axiological scale of society (traditions, relevant history, common law, geo-political placement, etc.).

Based on the above factor, we suggest that social order is *a normative combination of freedom and democracy that delivers a degree of individual self-respect/individual dignity*. Thus, the main features of social order are the following: (a) Social order is understandable in terms of norms or normative framework (more exactly, normative network) of society; (b) norms that ground social order are primary norms, that is, the basic principles of society; we consider such norms as the primary goods, in the sense suggested by Rawls [7]; (c) if social order is maintained inside maneuver room shown in Figure 3, some degree of individual self-respect is generated necessarily. It is not relevant at this point to develop a discussion regarding the moment inside the social contracting process when such a principle is chosen—either at the original position (as Nussbaum wants) or at the legislative stage (as Rawls demands).

### 3.5. An Abstract Typology of Society Based on Social Order

First, there are two marginal types of society: (a) Quasi-anomic society, and (b) supranomic society. Between these types there is, of course, ordinary society, that is, the *nomic society* (NS), which has an intermediate degree of social order, namely between the minimal and the maximal ones.

By *quasi-anomic society* (QANS) we mean a society in which the normative framework is minimal, referring only to the strictly necessary set of social interactions. A society cannot be absolutely anomic. There is always a set of rules/norms that work in any social group—that is why the polar category that refers to normative insufficiency is called a quasi-anomic society. In addition, as a rule, a quasi-anomic society does not have a formal (codified) normative framework but rather an implicit (tacit) one. Libertarians, for example, focus their argument on the lack of need for the state precisely based on the innate ability of individuals to develop, on the basis of their simple human nature, tacit “norms” of social interaction (the example of a narrow bridge over a river is famous). It is very important to remember that a quasi-anomic society is characterized by a very low degree of social sophistication, so it is specific to autarchic societies. From a theoretical point of view, however, this type of society certainly has the “right” to exist even if there are no such examples anywhere on the planet.

By *supranomic society* (SNS) we mean a society whose normative framework is maximum, or whose normative framework has two fundamental characteristics: (a) It regulates behaviors/actions that could be (and, actually are) regulated even by individuals involved in those behaviors/actions, and (b) it regulates beyond the level of regulation maintaining the criterion of regulating legitimacy. There is, of course, a threshold that should not be exceeded by the regulatory framework, such that societies remain standardly nomic. In fact, there are two categories of thresholds that should not be exceeded in the process of social norming: (1) The *competence threshold*: The principle of subsidiarity must work here—with no external regulation in which regulation is more effective as self-regulation, and (2) the *redundancy threshold*: The principle of sufficient reason must work here—with no further regulation of regulation. A certain situation can be called the “normative illegitimacy” of the state (although the regulations in question may, of course, be legal, i.e., they may be subject to constitutional rigors, as they may be, otherwise, illegal). There are level “0” regulations (constitutional regulation), level “1” regulations, also called primary legislation, followed by analytical levels of regulation: Including level “2” regulations, which regulates the application of level “1” regulations (so-called methodological or procedural rules, or secondary legislation); and level “3” regulations, which governs the application of regulatory level “2” (tertiary legislation), etc. The principle of subsidiarity (specific to decentralized societies, like all modern societies) requires that the regulatory decision be made where they exist simultaneously: Maximum competence, maximum interest, and maximum performative capacity (for example, in the framework of European Union legislation, this principle of subsidiarity works through the “division” of legislative powers between European and national regulatory institutions; there is also a “module” in which the two categories of regulatory institutions cooperate—see the Treaty on the Functioning of the European Union).

### 3.6. Social Entropy

Based on the concept of social order, we can now start to construct the concept of social entropy. Some relevant “ingredients” are already noted above, namely social entropy, which must verify the following:(i)It cannot be of a Thermodynamic type;(ii)It must be connected with social order;(iii)Its connection with social order must be inversely proportional;(iv)It must hold the connotations (either quantitatively or qualitatively) of the relationship between the homogeneity and heterogeneity of a system/process;(v)It is fundamentally grounded by normativity.

#### 3.6.1. Predictability vs. Novelty

The normative framework of society produces a degree of predictability regarding individual behaviors, analogous to the genetic code in Biology or to Luhmann’s concept of programs aimed at to “predicting” future communications. The more analytical the normative framework is, the more predictable the behaviors are. As predictability is an attribute of social order, the more analytical the normative framework is, the more ordered society will become. Thus, prima facie, a dictatorship provides the maximum “social order”. The full lack of norms is, from the standard concept of entropy, equivalent to the full coverage of norms, because, in both cases, homogeneity is at its highest level. However, predictability is equivalent with non-freedom, because predictability requires social necessity—that is, a full lack of free will. Thus, the second pillar of social order (freedom) is violated. This produces the need for a realm of non-normed social acts. In other words, a realm of novelty must be guaranteed by the social contract. Indeed, the “quantitative” split between the normed realm and the non-normed realm is an empirical and socio-historical issue, not a theoretical and logical one, as, qualitatively, both realms must co-exist. The necessity of a non-predictable realm inside the social field is a very important view that deserves to be further detailed:Some perturbations could emerge from inside society concerned, because norms, even as imperatives, can be violated. Thus, the normative framework’s functioning is, to some extent, random—despite the sanctions for violating norms, the cost of violation is acceptable [15];A well-ordered society [16] must face perturbations from its environment (either natural or societal), which no imaginable normative framework can technically take into account (and must not from a morally standpoint), so these perturbations must be offset based on free-will, thus, a non-normed social realm is required;Social evolution is conditioned by novelty. Full predictability simply provides a tautological replication of society, so a mutation-maker is required to provide such novelty—this mutation-maker is hosted by the aforementioned non-normed realm of the social system.

So far, we have obtained the first sufficient predicate of social entropy—the mix between the normed and non-normed realms of society. A logical consequence of the mix between the normed and non-normed realms of society is that, inside society, the system co-exists with the network. Indeed, while the normed realm acts as a system, the non-normed realm acts instead as a network. Thus, a well-ordered society must combine the system and the network. Fundamentalist contractarians, like John Rawls, understand a well-ordered society to mean society that is massively normatively ordered, while fundamentalist libertarians, like Hayek, understand the well-ordered society to be exactly the opposite. Clearly, a functional mix between the two accounts should be considered.

So far, we have introduced into just one of the three pillars of society: Freedom, which is objectified by the non-normed realm and, equivalently, by the network in social functioning. We must proceed further with the two remaining pillars. When we speak of the non-normed realm of the society, we understand society as an absolutely anomic realm, with the lack of operational norms, although its fundamental political principles are valid and in force for the entire society. In other words, society is regulated throughout by institutive principles and a certain realm is also operationally normed, while another can introduce its own operational norms which still verifying its institutive principles.

#### 3.6.2. Democracy and Order

Democracy is a way to make public decisions. To some extent, democracy can be logically derived from the concept of freedom (while the inverse derivation is rather impossible or at least ambiguous, as it could also lead to authoritarianism). Also, to a significant degree, democracy maintains (or defends, if required) freedom. Nonetheless, we insist on distinctly examining this pillar for social order, because of its relevance in guaranteeing freedom and for its role in splitting and conserving the split between the normed and non-normed realms of society. The ways in which democracy function, based on the account we develop in this paper, are disputable: (i) The higher the threshold at which a public decision is made, the greater the role of the minority in such decision making; (ii) the lower the threshold at which a public decision is made, the smaller the role of the majority in such decision making; (iii) there is a phenomenon that extends the minority’s option throughout society exactly because the majority strives to respect the minority’s option.

Despite these vulnerabilities (and others, as well) democracy will be retained as the third pillar of social order. It must be understood there is no break between the normed realm and the non-normed one. In fact, a continuum between the two realms exists: A necessary norm appears at the beginning and is then possibly codified (meanwhile, the future norm may already informally drive behavior); conversely, a norm that has been already codified can proves inadequate and become de-codified, although, it could persist, for a while, in actual behavior.

#### 3.6.3. Sketching Social Order Architecture

Based on the above considerations, we can conclude that social order is the result of the institutionalization and functioning of the three pillars: Self-respect, freedom, and democracy. A synoptic view of this result is provided in Figure 4. {The virtual state in Figure 4 addresses a hypothetical case in which the globally experienced pandemic of COVID-19 will reshape the world, sociality, and even the human condition. Social philosophers, psychologists, and politicians should stringently consider this.

As already implicitly understood, the three pillars of social order are lexicographically ranked from top to bottom as: Self-respect, freedom, and democracy—the SRFD social order (what we have called the DTI model).

Why does self-respect occupy the first place in our list? Why is it more important than freedom and, particularly, why does it not simply move people toward selfishness? Our answers are as follows. Additionally, it is easily proven that equal liberties for individuals (as claimed by Rawls, in his concept of the primary goods) cannot deliver equal self-respect for individuals. However, here subsists, in nuce, a more subtle issue: Like the concept of fairness/equity (which authorizes the inequality of means/resources in order to assure the equality of chances/outcomes), the concept of self-respect could authorize the inequality of liberties to assure the equality of self-respect thus providing a generous room for positive discrimination).

According to the Kantian account on the person, as well as most theories of social justice, the individual must always be considered an end and never (only) a mean for others’ ends; however, the highest end of an individual is his/her dignity or, likewise, his/her self-respect. Thus, self-respect is the maximum state that individuals can obtain within society, which places it at the top of our list;The logical priority of self-respect relative to freedom is offered by the derivability of freedom from self-respect. On the one hand, obtaining self-respect means that any individual must be treated as end. On the other hand, obtaining freedom means that one’s own purpose should not be established (or primarily imposed) by other individuals according to their own purposes. Consequently, freedom is a way through which self-respect is generated and conserved. This is why self-respect is positioned over freedom. Of course, alongside this assertion (which is an assertion about existence) a connected question immediately arises: Is freedom the only way to provide and defend self-respect? It is obvious that this question is unique. We do not explore this question in the present paper, but a good starting point might be the relationship between individual’s freedom and individual’s security (see, for example, the current debates against the COVID-19 pandemic).

The potential of self-respect to move an individual toward selfishness is the key element in justifying the logical priority of self-respect. We argue that the very definition of self-respect entails the inter-individual reciprocity. Indeed, to enjoy self-respect, an individual must be taken into consideration by others as being an end and only an end and never (only) a mean. Since all individuals must meet the same conditions for their own self-respect, one will not pass beyond pure self-respect, into selfishness, because, through reciprocity, that individual could be taken by others as a mean for their own purposes. Thus, self-respect, if applied to all of society, forbids its own deterioration into selfishness. Generally, the logical ranking of the three pillars of social order is inversely to their chronological (or operational) order. That is, democracy enacts a normative framework to provide freedom which, as we have shown, provides self-respect. Martha Nussbaum is among the philosophers who also consider self-respect to be of prime importance, at least from the point of view that public policy must ensure human capabilities for people functioning rather than the functioning of presumed such capabilities. There are, however, two paths to fool selfishness: Charity and benevolence. When these two behaviors, which are species of tolerance, are objectified, they can hide selfishness, under the mask of altruism; see, for a “genetical” altruism, Nussbaum’s approach for capabilities related to social justice [8].

Another question is relative to losses: Is self-respect lost when passing into freedom, and, conversely, is freedom lost when passing into democracy? Our position consists of the following points:
To lose some range of self-respect when passing into freedom is inevitable—as this is the price of reciprocal self-respect; thus, generally, the socialization process has a synergic effect called: “some losses of the natural state”;Likewise, when passing from freedom to democracy, some prerogatives of personal liberties are transferred to society’s representatives who are empowered to handle the transfer of people’s sovereignty. People’s sovereignty is transferred only as a use, not as a nominal “property”, which is always possessed by people. Democracy’s functioning requires that the minority accept the purposes proposed by the majority, so the minority are given purposes that they would not have proposed themselves, which would make democracy non-functional. There is a crucial issue here: The democratic imposition of an end that is not one’s own end, does not violates individual freedom, since such an end is publicly imposed, not privately (notably, freedom entails the lack of a private end of other to be imposed to an individual).Next, we address an issue regarding the other utilities that society aspires to obtain for individuals to remain stable and worthy. In other words, can really self-respect also include substantive utility (for example, mutual economic advantage) and evidential utility (for example, the justification of equality before the norm)? Briefly, we argue the following:
Reciprocal self-respect logically entails commutative justice, that is, the distribution of economic product within society, based on merit, so, substantive utility is causally entailed by the symbolic utility; for distributive justice, we do not distinguish between individuals with the full capacity for productivity and individuals without this full capability; thus, distributive justice also functions based on reciprocal self-respect;Equality before the norm is also a result of reciprocal self-respect, for freedom norms cannot engage in discriminations without affecting reciprocal self-respect; consequently, evidential utility is objectified by very symbolic utility, like in the case of substantive utility;Moreover, self-respect represents the core of the social justice, understood in its larger range, which Nussbaum maintains [8]. The main strand of argumentation is as follows:
Self-respect includes the dignity of human life, both of the individual concerned and of others, based on the necessary reciprocity;In turn, dignity obviously entails equality before the norms, equality of social opportunities (both economical and non-economical), and, generally, the rough equality for benefits provided by non-slavery social cooperation.

Finally, we address the relationship between self-respect (or, very suitable, freedom) and security. The state is taking over not only the organization of social cooperation, but also the defense against menaces that individuals cannot face themselves. Thus, alongside the price of reciprocity, individuals must also be charged with the price of security. Temporarily, reversibly, and punctually, some liberties (and, as a result, some ranges of self-respect) can be suspended (by the state only, that is, based on democracy) to prevent, guarantee, defend, and/or repair the effects of menaces that occur either predictably or accidentally; for example, the ongoing case of the COVID-19 pandemic.

### 3.7. Structure and Mechanism of Social Entropy

#### 3.7.1. The DTI Model of Social Order

As already suggested above, in this paper, we build a concept of social entropy starting from the concept of social order. Since, in society, evolution is an inextricable mix between objectivity/necessity and free will, social entropy must not be conceived through a simplistic analogy with social energy, social exergy, social temperature, etc. [1,3]. However, the connotation of the disorder degree deserves to be retained, albeit with an adjusted shape.

We argue that social entropy must not express the disorder per se, but simply a deviation (and, of course, the degree of such a deviation) of social order from the structure of the three pillars considered adequate for that society. We name the adequate structure of these three pillars, that is, their relative positions to each other, as DTI-structure. The DTI-structure can be characterized as follows:

DTI is a quasi-Pareto type model applied to three “individuals” in a system, namely: Self-respect, freedom, and democracy; we call this model the *DTI model of social order*; be aware that, in our conceptual construction of the Pareto-type model of social order, this model is not utilitarian at all. Rather, our proposal is of a procedural (or even functional) type, because the three “individuals” involved do not gain/lose any utility. In fact, mutually fitting these individuals aims to provide the maximum synergy (either inferential or emergence kinds) within society.

Although the DTI model is not utilitarian based on substantive-kind of utility, it is, however, quasi-utilitarian if symbolic utility is considered instead [17]. Thus, the DTI model aspires to its symbolic maximum utility;The problem is: How do the contributions of the three pillars move relative to each other in the total symbolic utility? We propose the following mechanism to account for this:
The measure of the total symbolic utility must be found in the first pillar—self-respect; any change in the second pillar—freedom—is acceptable if and only if self-respect in society at least is conserved. One can extract from this assertion a kind of principle of difference very similar to that of Rawls regarding income/wealth distribution, namely, *reducing liberties is acceptable if and only if self-respect in society as a whole is improved or, at least, conserved compared to any alternative dynamics of the liberties concerned*. For example, the limitations of mobility and the conditioning of direct social interactions generated by the COVID-19 pandemic (such as wearing a mask) across the world can and must be justified to determine if and to what extent such limitations and types of conditionings do not worsen the respect (the same applies for the case to war or natural calamities).However, any changes in the third pillar—democracy—are acceptable if that change at least conserves freedom *and* self-respect;Thus, for $\Delta SR_{i}$ the change occurs in the first pillar (self-respect); for $\Delta F_{i}$ the change occurs in the second pillar (freedom), and for $\Delta D_{i}$ the change occurs in the third pillar (democracy). This can be written as the following “true” logical relations:$$\left(\Delta F_{i}\right)\to \left[\left(SR_{i+1}\neq SR_{i}\right)⋀\left(\Delta SR_{i+1}\geq 0\right)\right],$$
where $\Delta SR_{i+1}=SR_{i+1}-SR_{i}$;
$$\left(\Delta D_{i}\right)\to \left\{\left[\left(F_{i+1}\neq F_{i}\right)⋀\left(\Delta F_{i+1}\geq 0\right)\right]⋀\left[\left(\Delta F_{i+1}\right)\to \left[\left(SR_{i+2}\neq SR_{i+1}\right)⋀\left(\Delta SR_{i+2}\geq 0\right)\right]\right]\right\})$$
The precise way in which self-respect (and its change) can be quantified is, currently, outside our interest.

#### 3.7.2. Assigning Social Entropy to Social Order

Next, we introduce our concept of social entropy. Basically, social entropy aims to model the oscillations of social order from the maximum of the DTI model (that is, from the maximum of symbolic utility represented by the reliable peak of self-respect). To be mentioned that such a maximum value is simply considered to be the previous level actually reached. A justified question arises here: Why do we need the concept of social entropy, when we already know its “difference to 1”, that is, the concept of social order? We believe this concept is required for the following reasons:It is almost impossible to measure self-respect dynamics and level, because, especially, self-respect is subjective and entails assessing the idiosyncratic characteristics of the individuals involved; in the same way used to measure well-being.Likewise, it is very difficult to measure (which can perturb the results of measurements) the losses of self-respect or freedom, when society necessarily and normatively passes from one pillar to another (see Figure 4);Thus, we require a set of proxies that can “translate” the deviations of social order from its desirable (the ideal, called DTI benchmark) levels and structures to worse ones;By measuring such proxies, with a relevant methodology, we can directly measure social entropy and, indirectly measure social order, which is, as shown, inversely indexed with social entropy dynamics.

Thus, we understand social entropy as a *quantitative measure of the deviation of self-respect from its possible maximum level* (that is, from the DTI peak). For this definition we must answer inherent questions such as: (i) What set of proxies can be measured and has the potential to express the assumed definition of social entropy? (ii) What is the mechanism that normatively works on these proxies? (iii) How are these proxies connected with the notions of the normative network, synergy, hubbing, and social justice? To minimize the space of the publication it was suggested that we postpone the analysis in anticipation of another possible intervention related to some concepts such as synergy or hubbing.

(i)Set of proxies to measure social entropy (the DIC list).The set of proxies used to measure social entropy is able to quantify the deviation of self-respect from its maximum. Notably, such a “maximum” is always empirical and never theoretical and depends upon the constitutional and legislative framework in force, as well as traditions, history, and other psychological and cultural factors of the society. Notably, we deliberately proceed in a reductionist way for reasons of simplicity, coherency, and transparency for understanding and implementing the proposal for social entropy; also, to respect Occam’s razor. The DIC list is provided below:(D) *Negative discrimination*, which addresses any kind of asymmetric treatment in society: Of genus, age, physical or mental dependency, race, nationality, and others of the same; for this issue, we must proceed with a great caution. For example, it is not a negative discrimination when a person with a mental disability is not employed at a workplace that requires superior thinking on the issues that have to be handled. We can say, with some precautions, that negative discrimination generally occurs among equals in physical and/or mental capacity. Since the syntagma “among equals” can be misleading, we suggest that “equality” must be framed by particular situations, and not just generally stated. For example, for a workplace that requires only that some documents to be read, a “normal” (i.e., able-bodied) person is equal to a wheelchaired person. Therefore, choosing the “normal” individual against the disabled one is a form of negative discrimination.(I) *Economic inequality*, which addresses asymmetries in economic product distribution that are not based on the asymmetries in effectively contributing to obtain that product, *when the individual potential to make a contribution is roughly equal*. It seems that economic inequality (not social inequality) can be best measured by the income and wealth distribution in society—for this point, we fully agree with Rawls [7];(C) *Corruption*, which addresses the situation in which people who are employed to handle public money claim and/or receive money either to perform what they have to do or to not perform what they are forbidden to do, no matter the degree or duration of such behavior. This description of corruption could be minimally expressed as: Privately capitalizing on one’s public position. Here, we provide the following distinct cases of corrupted behavior among public servants, or among contractual employee in public functions: Claiming or accepting to be distinctly paid. The Table 1 below synthesizes the “classes” of corruption:

The three proxies to measure social entropy can at most quantify the changes in social entropy (and, as shown, the changes in the related social order) but not the absolute level of such changes. Thus, we need an a priori assumption by which the changes in the proxy levels are translated into the changes in social entropy and, subsequently, into changes in social order. To this end, we introduce lexicographical ordering among the proxies in descending order: (D)–(I)–(C). This ranking indicates that the most important factor to increase social entropy is negative discrimination. The reason for this ranking is analogous to that of the above ranking of social order pillars: Both economic inequality and the corruption’s impact can be ¬translated” into negative discrimination, while the reverse path is not logically possible.

(ii)The mechanism to measure social entropy.

To describe an appropriate mechanism for measuring social entropy, we firstly need to understand the channels through which the three factors act both among themselves and on the three pillars of social order. To this end, the representation in Figure 5 synoptically shows the comprehensive mechanism through with social entropy functions.

In Figure 5, (+) means “generates an increase of”; (±) means “generates an ambiguous quantitative effect”; (↑) means “improves”; (↓) means “makes worse”; and (↑↓) means “causes an ambiguous effect regarding the progress of”.

Here, we will briefly deliver an assessment of the proposed mechanism to relate social order and social entropy to each other.

*Corruption* increases economic inequality via a hubbing effect—that is, those who already have more money, can corrupt, and will thus gain money this way, with which they will corrupt again, and so on; the hubbing effect is a type of positive feedback applied on the knot of a network [18]; moreover, corruption can directly increase negative discrimination, because it generates perturbation in equal access to social opportunities; at the same time, corruption negatively affects the democratic game of society, because it violates the range of approved laws in a democratic way;*Economic inequality* has an ambiguous impact on negative discrimination—its increasing can either increase negative discrimination through monetizing (of, for example, some filters for accessing to opportunities) or decrease negative discrimination by prompting charitable actions from favored individuals to the unfavored ones; also, increases in economic inequality will negatively affect freedom in society, because many social opportunities are, irremediably, conditioned by economic power (for example, education or healthcare);Increases in *negative discrimination* will worsen all social order pillars (self-respect, freedom, and democracy) for obvious reasons;Inside social order itself:*(d.1) Democracy* can have an ambiguous impact on freedom—for example, by rising above a “redline” in regulating social relationships or, conversely, decreasing below a “greenline” of de-regulating (or, likewise, avoiding regulations) social relationships. Thus, only keeping democracy between the “redline” and the “greenline” can yield a positive impact on freedom; instead, any improvements in democracy (through the “red-green” tunnel) will necessarily improve self-respect and, reciprocally, any improvements in self-respect will necessarily improve democracy. It logically seems that improving self-respect necessarily improves democracy within the “red-green” tunnel. Thus, only an independent (from self-respect) variation of democracy could place the latter outside of the “red-green” tunnel.*(d.2)* Improving *freedom* will, in turn, improve self-respect, while independently improving of self-respect will have an ambivalent impact on freedom; this is a very interesting case from a sociological and political perspective—improving self-respect could produce a normative narrowing of freedom. If such narrowing is, depending on the political power distribution inside society, captured by a part of society (either as the majority or the minority), then freedom could be narrowed.

### 3.8. Qualitative Analyses

The social entropy concept introduced above (the DTI model of social order together with the DIC model of social entropy) is an intuitionist model, since it is based on lists. Such lists, despite being logically grounded as internal rankings, are actually established based on intuitions regarding the essence of social order (self-respect), and the social entropy (negative discrimination). The common features of the two connected models are the norms—generally, the normative framework of the society, and more precisely, the normative network. Below, we engage in some qualitative analyses on social entropy as a DTI-DIC model.

#### 3.8.1. The Relevance of the Normative Network

All the three factors of social entropy (DIC factors) work in a decentralized way. We can presume that no society norms any of them by semes. Thus, for increasing social entropy society must function almost exclusively horizontally. It is true, on the other hand, that many spontaneous actions to increase social entropy can be allowed or “invited” by codified norms that are too permissive, enacted based on legislative incompetence or even knowingly. Equally, this requires social entropy dynamics to need the normative network. As shown before, a “normal” society holds a very large area of the normative network, from which in an extremely small area, state intervention is legitimate. Elsewhere, two of the present authors have formulated a theorem of separability regarding state intervention in society. Such an intervention, in a decentralized society, is legitimate if and only if the relevant phenomenon is not self-testable (for example, economic monopolization is not self-testable at all, while the market equilibrium price is fully self-testable). More precisely, state intervention is legitimate under two simultaneous conditions: (a) The phenomenon generates negative externalities for society as a whole, and (b) the phenomenon is dominated by positive feedbacks (that is, it necessarily enters a self-catalyzing process). Thus, if the general normative framework has sufficiently large meshes, then a sui generis normative network aimed at exploiting the meshes arises from nourishes itself, thus increasing social entropy. This is why our paper does not fully connect social entropy to normative framework, instead to the normative network. Notably, the compulsory character of the normative network inside society allows all social entropy factors to work, not only to the basic ones (e.g., corruption). Moreover, the occurrence of a normative network is a strong example of inference-type synergy (the type of synergy whose effect is predictable starting from the inter-connections model inside the social system (the alternative synergy type is emergent)).

#### 3.8.2. On the Connection between Social Entropy and Social Order

In the societies with a strongly narrowed normative network (for example, dictatorial societies), the three factors of social entropy are kept under control, but in a non-democratic way. This is why our paper does not discuss social entropy per se, but explores it together (causally, structurally, and functionally) with social order. In other words, social entropy simply does not linearly decrease with decreases in the level of its factors (symmetrically, it does not linearly increase with increases of its factors’ levels), so, its dynamics must be assessed together with social order dynamics. Here, the issue of thresholds arises. We use a graphical representation to highlight these dynamics (Figure 6).

#### 3.8.3. Social Entropy and Social Order’s Autopoieticity

Usually, autopoieticity approaches for society are focused on the social system as a whole, although some studies examine subsystems of the social system (especially economic ones). It would also be relevant to examine not only some subsystems, but some processes (or phenomena) that transversally shape and reshape the social system. In the present paper, this issue will be only tangentially touched upon (two of the present authors are currently publishing more on the matter). To this end, the sufficient “ingredients” of social order will be further established, based on their (causal) relationship with social entropy as explicated above.

The *genotype* of social order: The set of semes and memes working inside society—that is, the basic principles that regulate and ensure the self-respect of each citizen in society. The mutations in the genotype are, generally induced by social entropy dynamics, through the three socially entropic factors (DIC);The *phenotype* of social order: The normative framework, with its networked component. The self-replication of social order means the replication of its phenotype, also including errors of “transcription”;The *membrane* of social order: This is a constitutional item, so it is of an institutional nature;The *environment* of social order: Since social order does not exhaust the social system, the environment of social order is the social system’s general functioning, including social entropy phenomenology. The fact that social entropy is “moved” outside social order but maintained inside the social system and shaped as an environment of social order, is crucial in the autopoietic image of social order, for which social entropy acts as source of perturbations (including mutations exerted on the social order’s genotype);The *internal operations* of social order: Clearly, these internal operations are closed against the environment of social order (i.e., closed against social entropy). Although social order’s operations are often caused by social entropy dynamics (as Figure 5 suggests), social order’s phenotype replication is autonomous from social entropy (similar to the process for biological beings);The *external interactions* of social order: Such interactions occur within the entire social system and, especially, within the DIC factors. As a result, social order builds a specific niche whose most relevant area is located right inside social entropy’s phenomenology. This niche, together with the internal operations, leads to a sui generis co-evolution of the pair: Social order–social entropy. Social entropy has a determinant role in establishing social order autopoieticity. In fact, between social entropy and social order mutual fitness is produced and re-produced [19]. This pattern, this time regarding the relationship between human beings and the natural environment, is called CHANS (coupled human and natural systems) model, but it can be easily particularized into binomial social entropy–social order.The *self-reflexivity* of social order: Social order represents the most important engine for self-reflexivity, that is, a permanent and continuous examination of the degree (and propensity) of practically fitting the social values embedded in its genotype. Although social entropy is the result of citizens’ behavior, self-reflexivity is at a lower level, because economic (i.e., material) factors seem to be the “preferred” drivers.

#### 3.8.4. Social Entropy and Complexity

The issue of complexity is still debated. Many scientists still do not clearly discern between complicatedness and complexity. However, these are very different concepts. While complicatedness defines only a (great) difficulty in accurately describing the causalities and phenomenology of a process/system (this difficulty might disappear if we acquire further relevant knowledge, so complicatedness has a historical nature), complexity describes the principled impossibility to predict (and, implicitly, to causally explain) a phenomenology. This is why complexity can occur in a network only, although not necessarily any network is unpredictable (fully or partially). We argue that a network is unpredictable [20] in principle if and only if the process/system concerned is “endowed” with free will. As free will is assigned only to humans, complexity occurs and is forever maintained in the social network, that is, in the social entropy phenomenon. The logical negation of a complex is a simplex, as the logical negation of a complicated is a simple. Thus, a complex process/system can simultaneously be either simple or complicated, just as a simple process/system can be either simple or complicated. A conceptual dissociation between complicatedness and complexity is provided in Table 2. Some analysts, like Biggiero [20], argue that the degree of complexity in an entity can be determined by the predictability of the behavior within that entity, in an inversely proportional relationship.

#### 3.8.5. Social Entropy, Social Order, and Social Justice

We name this issue the “EOJ” social paradigm (that is, entropy-order-justice). We previously showed that social entropy and social order are linked in a co-evolutionary process. In a somewhat excessive metaphor, in a (Lotka-Volterra type of) co-evolution, social entropy would be the “predator”, while the social order would be the “prey”. However, it can be easily observed that both social entropy and social order are designed, defined, and interpreted from a social justice perspective (self-respect, freedom, economic inequality, negative discrimination). Thus, our proposal for the concept and phenomenology of the social entropy is largely grounded in social justice values and requirements. Thus, it is natural to combine these three factors and state that our model of social entropy is, in fact, an EOJ model or paradigm. Such grounding is required for the sociality of social entropy, which is opposed to the physicality of social entropy, most of the approaches in the literature).

## 4. Social Entropy-Based Typologies of Society

Our concept of social entropy has the potential to deliver a typology of societies based on their entropic class. We deliberately use medical terminology, given by the *IIHH* typology; (used for the suggestions provided in Figure 7), to create this typology of social entropy:Entropically *infected* societies: Societies whose DIC factors are maintained on the ascending side of the social entropy curve, between “null” level (in real societies, the DICs start from the tolerance threshold) and the inflection point;
This social entropy mark is, generally, specific to quasi-anomic societies;Such societies have already triggered the process of increasing social entropy, usually based on corruption, which produces economic inequality and/or negative discrimination. In this stage of the social entropy process, the hubbing effect is very active and productive, and (according to Figure 6) social entropy is accelerated; if a quantitative (i.e., numerical) measure must be used, it must increase marginal social entropy. Of course, on the concave part of the curve, marginal social entropy decreases;The aggregate outcome of the social entropy mark is, of course, the result of all the three factors in DIC in the convex trajectory of the social entropy curve, although there is a difference among them from perspective of both intensity and reversibility. Corruption is the most intensive and difficult to reverse, economic inequality is intensive and spiraled (it can increase or decrease but never regain a previous level reached), and negative discrimination is the least intensive and most easily reversible; that is, a hysteresis effect.Entropically *ill* societies: Societies whose DIC factors are maintained on the ascending side of the social entropy curve, namely between the inflexion point and the alarm threshold;
This social entropy mark is, generally, specific to nomic societies;The alarm threshold occurs when a social bifurcation point is reached where society could equally move towards social chaos (which will trigger social disintegrating) or towards a decrease in social entropy;However, the increase in social entropy is decelerated (i.e., the social entropy curve is concave);Since the social entropy of ill societies types is likely to merge within nomic societies, such societies have the potential to become healing and nomic societies, ill and quasi-anomic societies, or ill and supranomic societies;Entropically *healing* societies: Societies whose DIC factors are maintained on the descending side of the social entropy curve, namely between the alarm threshold and the inflection point;
This social entropy mark is, generally, specific to nomic societies, like with entropically ill societies;The tendency of social entropy is to decrease in an accelerated way (the social entropy curve remains concave);Depending on very small impulses, societies with this social entropy mark could remain as nomic societies and move into either ill societies or healthy ones, as well as towards supranomic societies;Entropically *healthy* societies: Societies whose DIC factors are maintained on the descending side of the social entropy curve, namely below the inflection point until reaching the “null” level (in real societies, up to the tolerance level);
This social entropy mark is, generally, specific to supranomic societies, with its dynamics much more strongly connected to norms of a seme type;Social entropy is decelerated and descending (the social entropy curve is convex);Social entropy is asymptotically approaching the tolerance level;There is a bifurcation point, that is, the society can pass from supranomic category towards a nomic one or from a healthy state of society towards healing state and even ill one. Here, an adverse effect arises, that could be called supra-nomicity, which aims at to reduce social entropy. This effect yields an increase in social entropy (specific threshold, however, remains to be found).

Of course, the shapes of the social entropy curve and those of the social order are simply illustrative, because their paths can be interrupted, reversed, or transformed any other shape, including bi- or tri-furcation points, according to concrete circumstances, both normative and praxiological. Figure 7 suggests some such logically possible contingencies.

## 5. Results and Conclusions

The concept of social entropy that we proposed is a political concept, not a technical one. There are some crucial reasons to proceed with this line of reasoning: (a) The social system (and, correlatively, the social order) are artifacts: They do not appear, disappear, or evolve and do not act else than through the human behavior. The concepts imported from Thermodynamics can play only a metaphorical (and suggestive) role in this area of study, but little else. As social theory has its own terms for its own concepts (for example, economic theory uses the term “heating”—economy heating—but in a metaphorical way only, because Economics has its own rigorous syntagma for such a phenomenon, namely a positive output-gap); (b) the social system (and, correlatively) the social contract are based not only on their substantive mutual advantage from social cooperation, but also (perhaps more so) on symbolic utility, i.e., on a moralized utility, that has nothing to do with the technical (often purely mechanical) approach afforded by an over-reliance on Thermodynamics;Moreover, our account of social entropy is located within a fully moralized society (at least, as an intellectual project, if not an actual one), specifically, within a society whose axiology is grounded on a comprehensive social justice account;Social entropy, as it is understood here, cannot be comprehended without social order. We showed, albeit briefly, that there is a co-evolution process between social entropy and social order, which is based on the autopoieticity of social order (also briefly sketched). This co-evolution process works as follows:When a change in the (social) order arises, some gaps (compared to previous order) appear, so a change in (social) entropy also arises (by either increasing or decreasing);To tentatively explain the changes in social entropy, two “assumptions” must be held: (1) An increase in social entropy entails an increase in the three pillars of the social entropy (that is, increases in one, two, or all three pillars). The increase of any of pillars (or their pairs) is not substitutable, nor can it be offset by other pillars increasing or decreasing. In essence, the process can be understood as follows:
b.1.The case of increasing social entropy—when social entropy increases, any of the social entropy pillars can increase, so at least one of the social order pillars will be negatively affected, so social order decreases. The same observation is valid for the social order pillars, like for the social entropy pillars, that is, they are not reciprocally substitutable or the subject of reciprocal off-sets;b.2.The case of decreasing social entropy—decreasing social entropy means that at least one social entropy pillar is decreasing. Thus, according to Figure 5, at least one pillar of social order is increasing, which produces social order improvement. This is the meaning in which we understand and use the concept of the co-movement (or co-evolution) of the two entities, of (social) order and (social) entropy.Social entropy is quantified by the DIC model: As negative discrimination, economic inequality, and corruption (in this descending order, based on the previous lexicographical procedure, founded on an intuitive argument), just as social order is quantified, by the DTI model: Self-respect, freedom, and democracy. We also provide, for both models, some logical arguments to strengthen and additionally justify the intuitionist method used for these rankings;The full functionality of social entropy needs a normative network inside the general normative framework of society. This “functional” prerequisite seems odd, but it is actually a logical (i.e., compulsory) consequence of the current (and desirable) decentralized model of social systems;In fact, the social entropy is a type of synergy (more precisely, inferential synergy) manifested by the generic social contract, and to a significant extent, increases around social hubs (this predictable effect has been named hubbing);Both social order and social entropy can deliver specific typologies of society: The first—three kinds of society: Quasi-anomic, nomic, and supranomic, and the later—four kinds of society: Entropically infected, entropically ill, entropically healing, and entropically healthy (IIHH typology).

Thus, paper’s results exhibit at least three novelties, particularly from a conceptual perspective:*Directly inferring of the social entropy from social structure and functioning*, more precisely from the concept of social order;Proposing of *social entropy structure*, as well as of *social order structure*;Proposing a *mechanism of the pair social entropy–social order*;Proposing a paired *typology of generic society*, from both social entropy and social order marks;Proposing a sui generis *principle of difference* related to the relationship between self-respect and freedom;Formulating the logical condition to *ensure the self-respect* when freedom or democracy changes, inside the social order.

The model proposed for connecting social entropy with social order, and for defining social entropy in relation with social order, can be evaluated as follows:(1)The model is logically grounded, but it is *institutionally functioning*, because it is deeply immersed into the normative framework (even more than that, into the normative network);(2)The model allows the *designing of public policy*, especially from the perspective of social justice, so that social entropy do not pass beyond the threshold that signifies a decrease in the social order (especially regarding the self-respect of the individual);(3)In fact, social entropy, as proposed in the paper, is both a true *predictor* for social order and a *leverage* for the latter, by handling the three pillars of social entropy;(4)More general, the model implicitly proposes a kind of *co-evolution* of the pair social entropy–social order;(5)The model invites researchers to move further by introducing some useful formalisms based on the complexity theory because of both presence of free will inside society and bifurcations points in evolution of pair social entropy–social order, as Figure 7 suggests.

## 6. Ways (Procedures) to Verify the Results

This paper’s results can be verified, like any results in the social sciences, only procedurally, not factually (i.e., not in a Popperian way) [21]. Much more so than in the natural field, the well-known Duhem–Quine thesis functions in the social field: Popper-ian testing cannot test the implicit hypotheses implied by hypotheses, thereby triggering an *ad infinitum* regress. The reason for the priority and relevance of this type of verification is grounded in the impossibility of the counter-factuality of the social field. This paper has already showed the ways in which the causal relationships among DIC’s factors and DTI’s factors arise. These causal relationships constitute the hypotheses that could, eventually, be procedurally falsified. Some additional suggestions regarding the verifications procedures of results are the following:(a)For the DIT model: The priority of self-respect against freedom can be verified (tested) by elaborating a logical model (procedure) of non-ambiguously deriving the necessity of freedom from self-respect. The same method can be used to sustain the priority of freedom against democracy—by deriving the democracy from freedom (independent of deriving democracy from self-respect);(b)For the DIC model: The lexicographical order of the components can be tested by elaborating a logical procedure through which negative discrimination directly implies economic inequality, economic inequality directly implies corruption, and negative discrimination indirectly implies corruption.

## 7. Directions for Further Research

Many issues have been left to be approached in the future, either by us or by other scientists with scientific interests in the matter. The most important of these issues are the following:(a)Are there threshold behaviors both for the three pillars of social order and the three pillars of social entropy? Could these two realms be connected by this threshold behavior?(b)Is there (and, if so, what is it) a link between the legality (i.e., constitutionality) in instituting low social entropy, on the one hand, and the legitimacy of the associated generated social order, on the other hand? Is it possible to design a causal (or, at least, correlational) model of such interference?(c)How could our proposed model of social entropy—DIT-DIC—be further developed to address special cases of institutionally integrated social systems, such as the European Union?(d)Could we design a general list of common social goals (or of “capabilities”, as Nussbaum names them) for social entropy to be institutionally constrained and tendentially approach its lowest level?

## Figures and Tables

**Figure 1 entropy-22-01051-f001:**
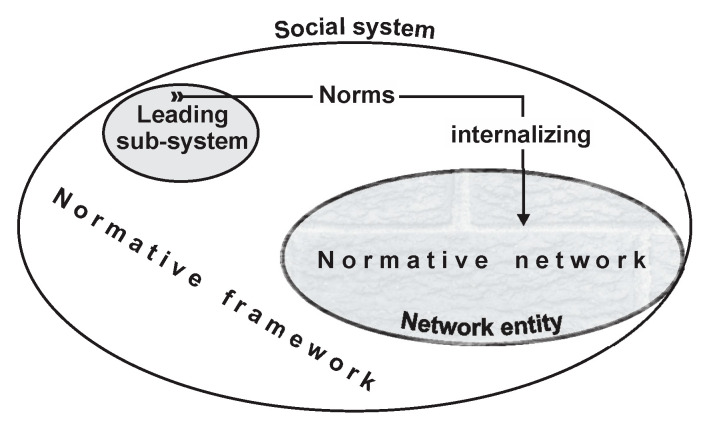
Logical mechanism of passing from normative framework to normative network.

**Figure 2 entropy-22-01051-f002:**
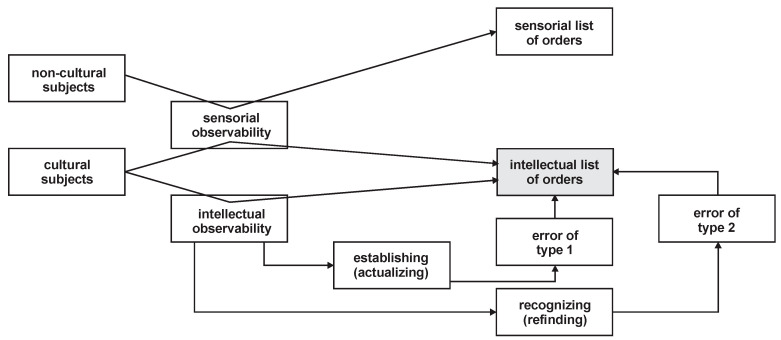
Dynamical formation of the intellectual list of orders.

**Figure 3 entropy-22-01051-f003:**
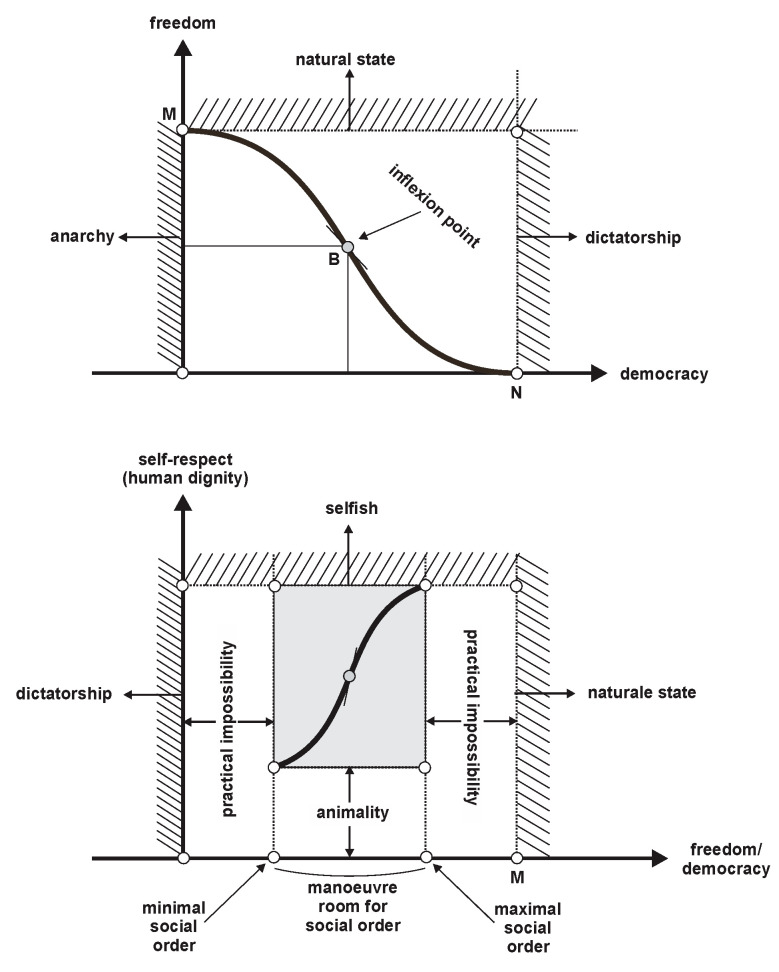
The causal and functional relationships among the three pillars of social order.

**Figure 4 entropy-22-01051-f004:**
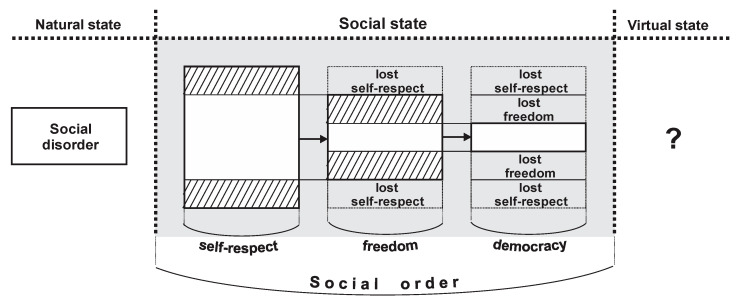
Necessary narrowing of the working space of social order pillars by lexicographically passing one to another.

**Figure 5 entropy-22-01051-f005:**
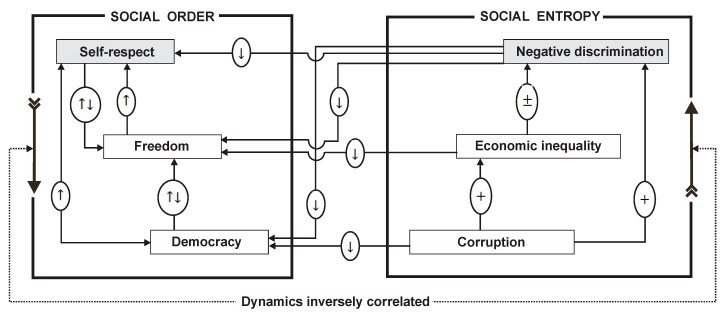
The comprehensive mechanism through which social entropy functions.

**Figure 6 entropy-22-01051-f006:**
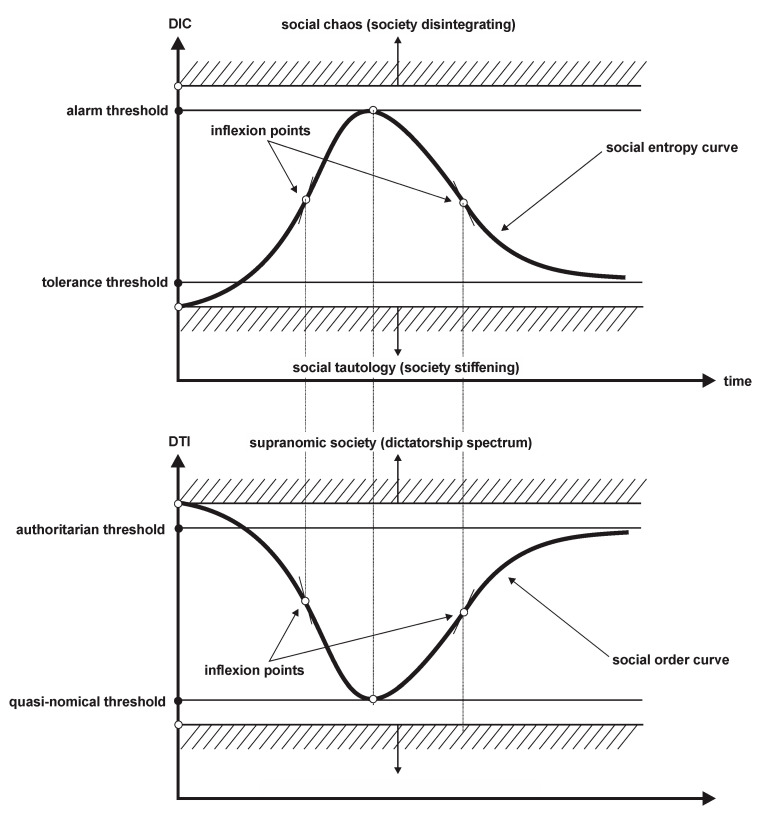
The non-linear dynamics of social entropy and the associated social order.

**Figure 7 entropy-22-01051-f007:**
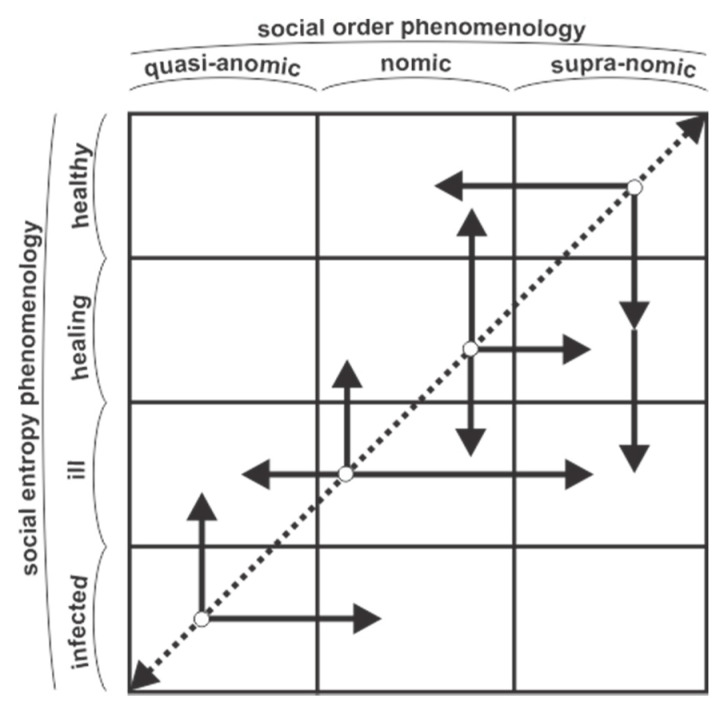
Imaginable contingent dynamics of the social entropy curve.

**Table 1 entropy-22-01051-t001:** Classes of corruption.

Case	Obliged to Do “x”:		Forbidden to Do “x”:	
	Actually Does “x”	Actually Does Not Do “x”	Actually Does “x”	Actually Does Not Do “x”
I	“legal” corruption			
II		“illegal” corruption		
III			“illegal” corruption	
IV				“legal” corruption

**Table 2 entropy-22-01051-t002:** The logical general relationship between complexity and complicatedness.

Aggregate		System		Network	
Simplex				Complex	
Simple	Complicated	Simple	Complicated	Simple	Complicated

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
