# Peer review of "Social Entropy and Normative Network"

_entropy, 2020, doi:10.3390/e22091051_

Round 1

Reviewer 1 Report

This paper introduces a new approach to social entropy and social order in terms of connecting to modern societies and social justice. The authors proposed DIC and DTI models to quantify the social entropy and social order. The end of the paper shows that the social order and social entropy can bring specific typologies of society.

However, there are some issues that need to be addressed.

  1. Although the graphical representation is sufficient to illustrate the proposed approach, the titles of the figures are not clear what the figure means. For example, the titles of Figure 3, and Figure 4 are the same. However, Figure 3 explains the example scenarios in the three pillars and Figure 4 shows the change of components when moving from one pillar to another. Additionally, descriptions of the figures should be more detailed.

  1. What is the related study of this paper? The author cited a few papers, and most of them are too old to reflect the latest research trends. It should be added to a more directly related study. Also, it would be nice to add references for general issues.

  1. The paper contains various and broad content since it approaches social order and social entropy with three pillars. Also, the logical organization is understandable, but the concepts described above are mentioned again later. Although the beginning of this paper gives an overall description of the content, it should be more clear about the overall organization of the paper in order to understand easily.

  1. The paper raises various research questions, but there is less clear verification process and result. In addition, the paper would really benefit from clarifying the main research question stressing the important aspects of this paper.

Author Response

First of all, the authors of the article „Social Entropy and Normative Network” would like to thank you for your efforts to improve the content of this paper. We will respond to your comments, point by point, as follows:

  1. Although the graphical representation is sufficient to illustrate the proposed approach, the titles of the figures are not clear what the figure means. For example, the titles of Figure 3, and Figure 4 are the same. However, Figure 3 explains the example scenarios in the three pillars and Figure 4 shows the change of components when moving from one pillar to another. Additionally, descriptions of the figures should be more detailed.

 Response 1:

  1. Although the graphical representation is sufficient to illustrate the proposed approach, (1. the titles of the figures are not clear what the figure means. For example, the titles of Figure 3, and Figure 4 are the same. However, Figure 3 explains the example scenarios in the three pillars and Figure 4 shows the change of components when moving from one pillar to another. (1.2. Additionally, descriptions of the figures should be more detailed.
  • The titles of Figures 3, and 4 have been modified. Thus, Figure 3 is now entitled „The causal and functional relationships among the three pillars of social order”, while Figure 4 is now entitled „Necessary narrowing of the working space of the social order pillars by lexicographically passing from one to another
  • Additional descriptions and explanations now accompany Figure 3 by lines 532-547; b) for Figure 4: its functioning is described in the lines 722-755

2. What is the related study of this paper? The author cited a few papers, and most of them are too old to reflect the latest research trends. It should be added to a more directly related study. Also, it would be nice to add references to general issues.

Response 2:

  1. What is the related study of this paper? The author cited a few papers, and most of them are too old to reflect the latest research trends. (1. It should be added to a more directly related study. Also, it would be nice (2.2. to add references for general issues.

2.1. We have added some supplementary critical evaluation regarding the most closed work to ours in the matter of social entropy (namely Bailey book, which is already mentioned in the paper, and which is the closest to our account regarding the social entropy) (see lines 145-160)

2.2. New references on the general issues belonging to the concept of entropy have been added and commented (see lines 992-996, 1016-1018, and 1146-1148)

  1. The paper contains various and broad content since it approaches social order and social entropy with three pillars. Also, the logical organization is understandable, but the concepts described above are mentioned again later. Although the beginning of this paper gives an overall description of the content, it should be more clear about the overall organization of the paper in order to understand easily.

Response 3:

  1. The paper contains various and broad content since it approaches social order and social entropy with three pillars. Also, the logical organization is understandable, (1. but the concepts described above are mentioned again later. Although the beginning of this paper gives an overall description of the content, (3.2. it should be more clear about the overall organization of the paper in order to understand easily.

3.1. The annoying repetition regarding the classical concept of entropy has been removed. Moreover, this well-known concept, in its technical understanding, has been totally transferred to Endnotes, which, however, because the template did not allow endnotes, that text was finally introduced into brackets in the main text.

3.2. We have introduced a new sub-section entitled „1.3. General organization of the paper”, which provides a clearer picture of the overall organization of the research in the paper (see lines 60-83)

  1. The paper raises various research questions, but there is less clear verification process and result. In addition, the paper would really benefit from clarifying the main research question stressing the important aspects of this paper.

Response 4:

  1. The paper raises various research questions, but there (1. is less clear verification process and result. In addition, the paper would really benefit from (4.2. clarifying the main research question stressing the important aspects of this paper.

4.1. The former point 5. The discussion has been renamed 5. Results and Conclusions (in fact, its content was exactly referring to results and conclusions of paper). Also, a new sub-section 6. Ways (procedures) to verify the results have been added to answer the recommendation (see lines 1144-1162)

4.2. The main research questions have been explicitly formulated (see lines 35-38)

Reviewer 2 Report

REVIEW PAPER ON SOCIAL ENTROPY

In this paper, the authors discuss the concept and the phenomenology of the social entropy from the point of view of the democracy and social justice values and requirements. Then, they propose a model of the social entropy in three dimensions namely, entropy-order-justice (EOJ)-model. According to their findings “… the paper gets the conclusion that the social entropy is an artefact, like the society as such, and it acts through the people behavior”.

The paper for sure is of some interest. However:

1). The length of the paper is too long. I would suggest the authors to shorten their discussions and the total length of the paper.

2). The structure of the paper is not appropriate as it is subdivided in many sections and subsections. I would also suggest the authors to remove the notations of bullet library. The same in the concluding remarks of the paper. It would be preferable to explain in full text their logic, methodology, and the main points of their findings and conclusions. For instance, subsection 2.1 exposes the concept of entropy, which is well-known. The three meanings of the entropy as described by the authors are repeated in subsection 3.3. I cannot see the reason of such a repetition. I suggest the authors to reduce this subsection in one paragraph (maximum two paragraphs) and provide some basic literature. Also, what is the necessity of subsection 2.7? Why not merge 2.6 and 2.7? Why not join 2.7.1 with 2.7.2? And so on.

3). I would suggest the authors to focus on the purpose of their research question. It seems that the focus of the paper is multiple, and this might introduce confusion to the reader. I would suggest the authors to precisely state their research question(s), provide a previous literature critical discussion and underline the contribution of their research.  

4). For instance, the authors are using the normative framework as a methodology tool in their analysis. I pass by the criticism of the model. According to the normative framework, three things could be done namely, (i) Identification and clarification of various goals of interest, (ii) Description of the trade-offs amongst these goals and (iii) Discussing the fundamental arguments about these goals while balancing the trade-offs. And very often, numerous normative goals are frequently in conflict with one another. However, the authors should explain and justify their choice.

5). (Lines 207-210). The authors want a signal (seme) to start from the leading sub-system are coming, both to itself and to the executive sub-system in a social network. However, in their research the possibility of “noise” is not taken into consideration, while the transmission of information (seme - signal) is probably full of noise in social networks. Then, in Line 219 they state: “... many useful memes are codified over time ...”. What it is meant by “useful”? Who recognize them? Is an institutionalized mechanism in charge?  

6). Line 274: “which is the second term in the equation social entropy”. Please explain.

7). Lines 984-986 (point 3). Please explain the co-movement of entropy with order. Same as Lines 34-35. From the brief discussion in Lines 109-118 it is not quite clear.

8). Please check English language with a specialist.

Author Response

First of all, the authors of the article „Social Entropy and Normative Network” would like to thank you for your efforts to improve the content of this paper. We will respond to your comments, point by point, as follows:

1). The length of the paper is too long. I would suggest the authors shorten their discussions and the total length of the paper.

 Response 1:

1). The length of the paper is too long. I would suggest the authors (1. to shorten their discussions and the total length of the paper.

  • Some phrases and parentheses have been removed when their value-added had too low signification. Also, some sub-sections have been removed when their contribution to core discussion seems to be also two low (for example, the sub-section 3.3, partially, and sub-section 3.6 „Social order and synergy”, totally, and other sub-sections have been put together into a single sub-section (see our answer at point 2.1. below)

Nonetheless, the valuable suggestions and recommendations to supplementary explain, develop or complete 35 pages (in the initial submission) now the paper has…34 pages, despite the numerous required (and justified) completions.

2). The structure of the paper is not appropriate as it is subdivided into many sections and subsections. I would also suggest the authors remove the notations of the bullet library. The same in the concluding remarks of the paper. It would be preferable to explain in full text their logic, methodology, and the main points of their findings and conclusions. For instance, subsection 2.1 exposes the concept of entropy, which is well-known. The three meanings of the entropy as described by the authors are repeated in subsection 3.3. I cannot see the reason for such repetition. I suggest the authors reduce this subsection in one paragraph (maximum two paragraphs) and provide some basic literature. Also, what is the necessity of subsection 2.7? Why not merge 2.6 and 2.7? Why not join 2.7.1 with 2.7.2? And so on.

Response 2:

2). (2.1. The structure of the paper is not appropriate as it is subdivided in many sections and subsections. I would also suggest the authors (2.2. to remove the notations of the bullet library. (2.3. The same in the concluding remarks of the paper. (2.4. It would be preferable to explain in full text their logic, methodology, and the main points of their findings and conclusions. (2.5. For instance, subsection 2.1 exposes the concept of entropy, which is well-known. (2.6. The three meanings of the entropy as described by the authors are repeated in subsection 3.3. I cannot see the reason for such repetition. I suggest the authors reduce this subsection in one paragraph (maximum two paragraphs) and (2.7. provide some basic literature. (2.8. Also, what is the necessity of subsection 2.7? Why not merge 2.6 and 2.7? Why not join 2.7.1 with 2.7.2? And so on.

  • A number of sub-sections or paragraphs have been abolished, taking into account both your suggestions and their concrete content reviewed. See below, also, the particular changes are done for 2.2 -2.6, and 2.8
    • The bullets have been throughout replaced by usual numbering (either numbers or letters)
    • Idem at 2.2.
    • The following actualizations have been made:
  • lines 35-38: distinctly formulation of the purpose of research
  • lines 27-29: explicitly formulating of a third aim of the research (which is, in fact, of course, performed in the study)
  • lines 53-54: introducing a new methodological peculiarity of the research (letter e.)
  • lines 60-83: a new sub-section, named 3. General organization of the paper, where, alongside the logic and methodology of the paper (which already are done), a „map” regarding the research organizing and basic results intended/obtained are presented in short and in their sequences
    • The conceptual description of the concept of entropy in paragraph 2.1 (which, indeed, are well-known by the publication’s usual readers), has been moved to endnotes, for the case in which one would want to see which is the understanding of the author’s regarding the basic concept of entropy (which could be of help for further pursuing of paper account developments). However, because the template did not allow endnotes that text was finally introduced into brackets in the main text
    • Sub-section 3.3. has been removed
    • New references, with short adequate comments in the light of the paper’s topic, has been added:
  • on the general issues belonging to the concept of entropy (see lines 992-996, 1016-1018, and 1146-1148)
  • on the social entropy issue (see lines 145-160)
    • Sub-sections 2.6 and 2.7 has been integrated into a single sub-section and renamed as „Norms Fetality and Normative Network”. The former paragraphs 2.7.1. and 2.7.2. have, so, disappeared as such. Regarding the paragraph on the necessity of norms (from the beginning of the former sub-section 2.7.), we kindly ask the reviewer to observe that the allegations held are not those commonly known, but they are made (only) and strictly confined on the issues which are of greatest interest for further considerations (social order and social entropy, respectively). So, we would desire, if agreed, to maintain those few propositions done in the initial text.

3). I would suggest the authors focus on the purpose of their research question. It seems that the focus of the paper is multiple, and this might introduce confusion to the reader. I would suggest the authors to precisely state their research question(s), provide a previous literature critical discussion and underline the contribution of their research.  

Response 3:

3). I would suggest the authors (3.1. to focus on the purpose of their research question. It seems that the focus of the paper is multiple, and this might introduce confusion to the reader. (3.2. I would suggest the authors to precisely state their research question(s), (3.3. provide a previous literature critical discussion and (3.4. underline the contribution of their research.  

  • The sub-section regarding the synergy has been removed
    • Lines 35-38: the questions which the paper is aimed at to answer are explicitly formulated
    • The almost all approaches of the social entropy are based, in line of Shannon’s opening, on the information theory, (including the revolutionary theory of the social system of Luhmann), when they are not directly related to Thermodynamics, which are too far from our approach based on the social justice. Moreover, it is proved (see, here, for example, Rudolf Arnheim, with his entropy paradox, caused by the informational basis of the entropy, namely: „total disorder provides a maximum of information; and since information is measured by order, the maximum of order is conveyed by a maximum of disorder”– Entropy and Art, University of California Press, Berkeley, 1971, p.4 – apud Philip J. Davis, Entropy and Society: Can the Physical/Mathematical Notions of Entropy. Be Usefully Imported into the Social Sphere?, Journal of Humanistic Mathematics, vol. 1, issue 1, 2011). Other approaches are focused on a topological (as well as on Mechanical Statistics) – for example, Christian Zingg, Giona Casiraghi , Giacomo Vaccario , and Frank Schweitzer, in What Is the Entropy of a Social Organization?, published by Entropy, on September 14, 2019), which is also diametrally opposed to our solution. The only proposal we know, and which has some (quite vague) tangencies with our suggestion in the social entropy matter, is the work of Bailey (cited in the paper). So, we have developed, in a little extent, and in a critical way, as recommended, Bailey’s account (see lines 145-160)
    • Lines 84-93: introducing of a new sub-section named 4. Main own contributions of the research, which summarizes the seven contributions to the topic of social entropy we think we have brought in our paper

4). For instance, the authors are using the normative framework as a methodology tool in their analysis. I pass by the criticism of the model. According to the normative framework, three things could be done namely, (i) Identification and clarification of various goals of interest, (ii) Description of the trade-offs amongst these goals and (iii) Discussing the fundamental arguments about these goals while balancing the trade-offs. And very often, numerous normative goals are frequently in conflict with one another. However, the authors should explain and justify their choice.

Response 4:

4). For instance, the authors are using the normative framework as a methodology tool in their analysis. I pass by the criticism of the model. According to the normative framework, three things could be done namely, (i) Identification and clarification of various goals of interest, (ii) Description of the trade-offs amongst these goals and (iii) Discussing the fundamental arguments about these goals while balancing the trade-offs. And very often, numerous normative goals are frequently in conflict with one another. However, (4. the authors should explain and justify their choice.

  1. The reviewer asks us regarding the basis on which we have chosen the normative framework as a methodological tool in delivering the analysis in the paper. Our main „defence” addresses, of course, the fact that we want to relate the social entropy to the social order. Thus, although we agree with the theoretical need to clarify the three aspects triggered by the concept of normative framework, in the perspective of our research topic and purpose, we would prefer to take such a normative framework as a given (that is, to consider the three mentioned issue as already balanced by the social contract agreement). The gaps of actual social order from the social order „guaranteed” by the normative framework is the only object of social entropy measuring.

5). (Lines 207-210). The authors want a signal (seme) to start from the leading sub-system are coming, both to itself and the executive sub-system in a social network. However, in their research, the possibility of “noise” is not taken into consideration, while the transmission of information (seme - signal) is probably full of noise in social networks. Then, in Line 219 they state: “... many useful memes are codified over time ...”. What it is meant by “useful”? Who recognize them? Is an institutionalized mechanism in charge?

 Response 5:

5). (Lines 207-210). The authors want a signal (seme) to start from the leading sub-system are coming, both to itself and to the executive sub-system in a social network. However, in their research (5.1. the possibility of “noise” is not taken into consideration, while the transmission of information (seme - signal) is probably full of noise in social networks. Then, in Line 219 they state: “... many useful memes are codified over time ...”. (5.2. What it is meant by “useful”? Who recognize them? (5.3. Is an institutionalized mechanism in charge?  

  • In our account in the matter, the semes, which are codified norms (that is, it is mandatory to comply them) are not bearers of noise at all, either at least because they are publicly enacted (being publicly enacted means, inter alia, transparently and informationally access to them without transaction costs), so the semes seem to be „clean” or „free” from noise. In a (probable, not too appropriate) terminology of microeconomics, the semes „market” is informationally efficient. However, from the point of view of fetality, it could exist some noise, but related to the non-fetality (and, more, to the abundance of such a non-fetality) of norms as semes (see added lines 306-307)
  • This is a great (and difficult) question. In our opinion, after the initial stage of the social contract „signing”, the following stages (for example, in Rawlsian chronology), should simply observe and codify the memes which have become in force by the „invisible hand” mechanism. So the memes usefulness is, in fact, a predicate which no one, except the impersonal, general, and massive using or inventing them can manage. However, the leading sub-system must make such a codifying, when a threshold of memes in force is overcome. In such a context, we indeed appreciate that some additional comments on the point are needed (see lines 277-280)
  • In the additional comments, according to the above point 5.2., such a mechanism is sketched (see lines 282-285) as belonging to the auto-poieticity of the society

6). Line 274: “which is the second term in the equation social entropy”. Please explain.

 Response 6:

6). Line 274: (6. “which is the second term in the equation social entropy”. Please explain.

  1. Our expressing seem indeed be unclear here. In the equation (Nota bene: of course, the term „equation” is taken here in a rather metaphorical way) social entropy – normative network, the first term is the social entropy, and the second one is the normative network which, according to Figure 1, is the result of combining the normative framework with the fetality of norms (i.e., of internalizing or, if wanted, legitimizing of externally issued norms). In lines 327-328 we have written with italics the „equation”, in order to make visible its two terms (to left is the first term, and to right is the second term).

7). Lines 984-986 (point 3). Please explain the co-movement of entropy with the order. Same as Lines 34-35. From the brief discussion in Lines 109-118 it is not quite clear.

Response 7:

7). Lines 984-986 (point 3). Please (7.1. explain the co-movement of entropy with the order. (7.2. Same as Lines 34-35. From (7.3. the brief discussion in Lines 109-118 it is not quite clear.

  • The co-movement is understood as the standard concept of co-evolution, that is: when a change in the (social) order arises, some gaps compared with the previous order appear, so a change in the (social) entropy arises too (by either its increasing or decreasing); the reciprocal connection is more laborious to be described (and it is even less obvious). To go into a tentative explanation, two „assumptions” must be held: 1) an increase of social entropy means an increase in the three pillars of the social entropy (that is, in one, in two, or all three pillars. To be observed that the increase of any of them or any pairs of them is not substitutable, nor can be off-set by other pillars increasing or decreasing). In essence, the process could be understood as happening as follows (see lines 1109-1126):
  1. the case of social entropy increasing – when social entropy increase, any, or more of the social entropy pillars increase, so at least one of the social order pillars are negatively affected, so the social order as such is decreasing (the same observation is valid regarding the social order pillars, like for the social entropy pillars, that is, they are not reciprocally substitutable or subject of reciprocal off-sets);
  2. the case of social entropy decreasing – the social entropy decreasing means at least a social entropy pillar is decreasing, so, according to Figure 5, at least one pillar of the social order is increasing, which comes to the social order improvement, as such. This is the meaning in which we understand and use the concept of co-movement (or co-evolution) of the two entities, namely (social) order and (social) entropy. In the revised version of the paper, additional clarifications are provided
    • Indeed, the phrase „In addition to it, an institutional look is also focused on the concepts describing the social order as well as on the ones describing the social entropy.” was no covered or taken over in the further approaches. Consequently, it has been removed
    • The mentioned paragraph has been removed – indeed, it's valued added to the core discussion had, any case, too less relevance. Moreover, it seemed to be somewhat a little bit far from the main strand of our interest in argumentation

8). Please check English language with a specialist.

Response 8:

The authors turned to the English language services of the Entropy Journal.

https://susy.mdpi.com/user/pre_english/back/22154

Round 2

Reviewer 1 Report

The novelty of this paper seems still poor. 

Also, more importantly, there must be careful discussion on evaluating the proposed modeling. 

Author Response

The authors of the paper „ Social Entropy and Normative Network” would like to thank you for carefully and thorough reading of this article, for the thoughtful comments and valuable suggestions, which improve the quality of our scientific article, and that we will take into account in our future research directions.

We have incorporated all of them within the paper and they are visible within response there where we specified the lines.

Our responses follow:

1. The novelty of this paper seems still poor. 

 Response 1:

  • The novelty of this paper seems still poor.
  • The novelty of the paper is provided by at least six results obtained in the research in case:
  1. The directly inferring of the social entropy from social structure and functioning, more precisely from the concept of social order and (which is a supplementary novelty) on social justice;

Comment: Social entropy is not anymore built as a (more or less) ingenious analogy with the Thermodynamic entropy imported from Statistical Mechanics into the social field, but from the normative network as legitimate normative general framework, which generates the social order, which, in turn, ensure the social justice of the society.

  1. Proposing of social entropy structure, as well as of social order structure;

Comment: Both social entropy and social order are defined (every) as containing three pillars (self-respect, freedom, and democracy – for social order; negative discrimination, economic inequality, and corruption – for social entropy). In fact, thus, the paper put before the scientific community a (polemical, of course, but possibly productive) new topic: the two structures evoked.

  1. Proposing a mechanism of the pair social entropy – social order;

Comment: The paper bindingly links the social entropy to the social order, showing that social order could be „measured” by intermediation of a more easily measuring of the social entropy, through the three proposed pillars for the latter.

  1. Proposing a paired typology of generic society, from both social entropy and social order marks

Comment: The paper proposes three sorts of society, from the point of view of the social order mark (quasi-anomic, nomic, and supranomic), respectively four sorts of society, from the point of view of social entropy mark (entropically infected, entropically ill, entropically healing, and entropically healthy)

  1. Proposing a sui generis principle of difference related to the relationship between self-respect and freedom

Comment: reducing liberties is acceptable if and only if self-respect in society as a whole is improved or, at least, conserved compared to any alternative dynamics of the liberties concerned

  1. Formulating the logical condition to ensure the self-respect when freedom or democracy changes, inside the social order

Comment: In fact, such a logical condition to be verified constitutes both a predictor for the social order sustainability and a fine tuning adjustor for that sustainability

According to the suggestion, the headings of the above described novelty are now introduced in the Results and conclusions section of the paper (see lines 1142-1151)

  1. Also, more importantly, there must be careful discussion on evaluating the proposed modelling.

Response 2:

  • Also, more importantly, there must be careful discussion on evaluating the proposed modelling. 
  • The core of the model proposed for the social entropy in its relationships with the normative network (that is, with the social order) is included in Figures 5, and 8 respectively. Also, after Figure 5, the paper provides a principled examination of the logical functioning of this model.

According to the suggestion, a supplementary evaluation of the proposed modelling is now introduced, also in the Results and conclusions section of the paper (see lines 1152-1166).

Reviewer 2 Report

Dear Authors,

Thank you for the new version of your manuscript. Even if the paper is still too long in length, I think that now its purpose is clear.

Author Response

The authors of the paper „ Social Entropy and Normative Network” would like to thank you for carefully and thorough reading of this article, for the thoughtful comments and valuable suggestions, which improve the quality of our scientific article, and that we will take into account in our future research directions.

Our response follows:

  1. Thank you for the new version of your manuscript. Even if the paper is still too long in length, I think that now its purpose is clear.

 Response 1:

Thank you, in turn, for your comments and suggestions. They were of great usefulness for us and contributed to a serious improvement of the paper.
